# Impaired mitochondrial oxidative metabolism in skeletal progenitor cells leads to musculoskeletal disintegration

Chujiao Lin[1], Qiyuan Yang[2,12], Dongsheng Guo[3,12], Jun Xie [4,5,6], Yeon-Suk Yang[1], Sachin Chaugule [1], Ngoc DeSouza[1], Won-Taek Oh [1], Rui Li [2], Zhihao Chen[1], Aijaz A. John[1], Qiang Qiu, Lihua Julie Zhu [2], Matthew B. Greenblatt [7,8], Sankar Ghosh [9], Shaoguang Li [1], Guangping Gao [4,5,6,10], Cole Haynes [2], Charles P. Emerson[3,11] & Jae-Hyuck Shim [1,4,10] ✉

Although skeletal progenitors provide a reservoir for bone-forming osteoblasts, the major energy source for their osteogenesis remains unclear. Here, we demonstrate a requirement for mitochondrial oxidative phosphorylation in the osteogenic commitment and differentiation of skeletal progenitors. Deletion of Evolutionarily Conserved Signaling Intermediate in Toll pathways (ECSIT) in skeletal progenitors hinders bone formation and regeneration, resulting in skeletal deformity, defects in the bone marrow niche and spontaneous fractures followed by persistent nonunion. Upon skeletal fracture, Ecsit-deficient skeletal progenitors migrate to adjacent skeletal muscle causing muscle atrophy. These phenotypes are intrinsic to ECSIT function in skeletal progenitors, as little skeletal abnormalities were observed in mice lacking Ecsit in committed osteoprogenitors or mature osteoblasts. Mechanistically, Ecsit deletion in skeletal progenitors impairs mitochondrial complex assembly and mitochondrial oxidative phosphorylation and elevates glycolysis. ECSIT-associated skeletal phenotypes were reversed by in vivo reconstitution with wild-type ECSIT expression, but not a mutant displaying defective mitochondrial localization. Collectively, these findings identify mitochondrial oxidative phosphorylation as the prominent energy-driving force for osteogenesis of skeletal progenitors, governing musculoskeletal integrity.

Mitochondrial diseases are associated with decreased energy production in organs with high energy requirements and are mainly due to defects in the oxidative phosphorylation (OXPHOS) machinery[1–4]. Releasing energy through OXPHOS requires the orchestrated action of five multi-heteromeric enzyme complexes (complexes I–V), located in the inner mitochondrial membrane[5]. Mitochondrial complex 1 (CI) is the largest complex comprised of at least 45 different proteins and is the primary entry point that generates the proton motive force for

[1]Department of Medicine/Division of Rheumatology, UMass Chan Medical School, Worcester, MA, USA. [2]Department of Molecular, Cell and Cancer Biology, UMass Chan Medical School, Worcester, MA, USA. [3]Department of Neurology, UMass Chan Medical School, Worcester, MA, USA. [4]Horae Gene Therapy Center, UMass Chan Medical School, Worcester, MA, USA. [5]Department of Microbiology and Physiological Systems, UMass Chan Medical School, Worcester, MA, USA. [6]Viral Vector Core, UMass Chan Medical School, Worcester, MA, USA. [7]Department of Pathology and Laboratory Medicine, Weill Cornell Medical College, Cornell University, New York, NY, USA. [8]Research Divisions, Hospital for Special Surgery, New York, NY, USA. [9]Department of Microbiology and Immunology, Columbia University Vagelos College of Physicians and Surgeons, New York, NY, USA. [10]Li Weibo Institute for Rare Diseases Research, UMass Chan Medical School, Worcester, MA, USA. [11]Wellstone Muscular Dystrophy Program, UMass Chan Medical School, Worcester, MA, USA. [12]These authors contributed equally: Qiyuan Yang, Dongsheng Guo. ✉e-mail: jaehyuck.shim@umassmed.edu

electrons into the electron transport chain driving ATP production[6–8]. CI-deficiency is the most frequent cause of defects of the OXPHOS system that cause multisystem dysfunction[9–13], including skeletal disorders[14,15].

Emerging epidemiological evidence[16–18] and data from animal models[3,19] suggest an association of mitochondrial dysfunction and high oxidative stress with skeletal disorders, including a clinically significant decrease in bone strength, reduced bone mineral density, premature aging of bone or a high risk or fragility fractures. For example, Kearns Sayre Syndrome (KSS), a rare mitochondrial DNA deletion syndrome, shows clinical manifestations of muscle weakness and wasting, accompanied with multiple bone deformities and severe fractures, causing loss of the ability to walk and early lethality[20]. In addition, mitochondrial respiratory chain deficiency is related to spontaneous bone fractures and myopathy[21]. However, it is unclear how mitochondrial defects contribute to skeletal dysfunction associated with myopathy.

Glycolysis is the primary metabolic process that generates ATP for energy and intermediate metabolites, including active lipids, nucleotides, and amino acids, for biosynthesis[22,23]. It serves as the major energy and carbon source for committed osteoblast-lineage cells during osteogenic differentiation[22,24–26]. In particular, metabolic intermediates fuel the synthesis of amino acids to produce extracellular matrix proteins and hydroxyapatite in bone[22]. In addition to glycolysis, glutamine metabolism also provides osteoblast progenitors with an additional energy source as well as a carbon and nitrogen source[26]. Glutamine is converted to citrate through the tricarboxylic acid (TCA) cycle in the mitochondria, producing ATP through the OXPHOS pathway, while providing metabolic intermediates for the synthesis of amino acids, nucleotides, glutathione, and hexoamine[27]. Recent studies demonstrated an in vivo role for glutamine and glucose metabolism in skeletal progenitors using conditional deletion of the enzyme glutaminase (Gls) and the glucose transporter Glut1, respectively. Gls-deficiency in Prx1+ skeletal progenitors impaired,cell proliferation and osteogenesis during bone remodeling while Glut1-deficiency in Prx1+ skeletal progenitors abolished chondrogenesis, not osteogenesis, during skeletal development[28]. Notably, glutamine is likely to serve as major substrates for biosynthetic and antioxidant demands, not an energetic substrate, in skeletal progenitors since GLS inhibition did not induce any energetic alteration[29]. Thus, other energy metabolic pathways controlling osteogenesis of skeletal progenitors remain to be elucidated.

To investigate in vivo roles for mitochondrial oxidative metabolism in musculoskeletal system, we developed a mouse model for mitochondrial CI deficiency by conditionally deleting Ecsit in multiple skeletal and muscle cells. Herein, we highlight the importance of mitochondrial OXPHOS in determining the osteogenic commitment and differentiation of skeletal progenitors as a critical rheostat of skeletal and muscular development. Notably, restoration of impaired OXPHOS pathway using recombinant adeno-associated virus (rAAV) may represent an untapped therapeutic avenue for mitochondrial disorders in the musculoskeletal system.

## Results

### Ecsit-deficiency in skeletal progenitors impairs early skeletal development

The CI intermediate assembly (MCIA) complex, containing the core subunits (ECSIT, NDUFAF1, ACAD9, and TMEM126B) is required for the formation of mitochondrial complex I (CI)[6,30–33]. Among these subunits, ECSIT is highly expressed in skeletal cells, including endosteal osteoblasts, osteocytes, periosteal cells, and osteoclasts (Fig. 1a), and its deficiency reduces the expression of other MCIA and CI components[31–33]. Mitochondrial localization of ECSIT in human bone marrow-derived stromal cells (BMSCs) and mouse osteoblasts residing on the surface of trabecular and cortical bones was also confirmed by

immunofluorescence and subcellular fractionation analyses (Fig. 1b, c). The role of mitochondrial activity in bone formation was examined in vivo using mice with targeted deletion of Ecsit in skeletal progenitors (Ecsit^Prx1). Deletion efficiency of Ecsit in these cells was confirmed by fluorescence microscopy and immunoblotting analyses (Supplemental Fig. 1a, Fig. 2c). Ecsit^Prx1 mice developed limb deformities along with spontaneous fractures in long bones right after birth (Fig. 1d, right). At postnatal day 21 (P21), they displayed severe osteoporosis and delayed formation of primary and secondary ossification centers in long bones (Fig. 1d, e, Supplemental Fig. 1b, c). Severe hypomineralization and delayed formation of primary ossification centers was also evident in Ecsit^Prx1 embryos and neonates (Fig. 1f, Supplemental Fig. 1e, f, Supplemental Table 1). Consistent with immunofluorescence analysis showing nearly complete absence of osteocalcin (OCN)-expressing mature osteoblasts in the femur of P21 Ecsit^Prx1 mice (Fig. 1f, right), cKO Prx1+ skeletal progenitors, isolated from P0 Ecsit^Prx1 hindlimbs as Prx1+CD45-TIE2-TER119- cells, failed to differentiate into mature osteoblasts (Fig. 1g). Of note, the bone marrow niche function of Ecsit^Prx1 skeletal cells was also disrupted, as shown by a significant reduction in total bone marrow cell numbers and a low frequency of hematopoietic stem cells (HSCs), B cells, macrophages, granulocytes, and T cells (Fig. 1h, Supplemental Fig. 2). These results suggest that ECSIT is essential for osteogenesis of Prx1+ skeletal progenitors and for the formation of bone marrow HSC niche. Notably, GLUT1-mediated glucose metabolism mediates chondrogenesis, not osteogenesis of Prx1+ skeletal progenitors[28] while GLS-mediated glutamine metabolism in Prx1+ skeletal progenitors is dispensable for skeletal development[29]. Thus, ECSIT-mediated mitochondrial regulation, not GLUT1-mediated glucose metabolism or GLS-mediated glutamine metabolism, plays a critical role in the osteogenic activity of Prx1-lineage skeletal progenitors.

Prx1-lineage skeletal cells are major cell populations (Fig. 1i) that initiate callus formation and vascularization for successful bone fracture repair[34–36]. Ecsit deletion in Prx1+ skeletal progenitors resulted in nearly complete arrest of cortical bone and periosteal formation despite expansion of Ecist-deficient periosteal cells (Fig. 1j, Supplemental Fig. 1f). Accordingly, Ecsit^Prx1 femurs show spontaneous fractures occurring after birth. The resulting fracture sites displayed persistent nonunion with a lack of callus mineralization and mature osteoblasts and the persistence of an unmineralized fibrous callus with cartilaginous elements (Fig. 1j, k, Supplemental Fig. 1d). To define where ECSIT is required in the osteoblast maturation process, skeletal cell populations were isolated from P0 Ecsit^Prx1 limbs and subjected to flow cytometry[37], demonstrating an increased frequency of skeletal stem cells (SSCs) and pre-bone, cartilage, stromal progenitors (pre-BCSPs) and a decreased frequency of BCSPs (Fig. 1l, m, Supplemental Fig. 3). Thus, Ecsit deletion in Prx1+ skeletal progenitors impairs the differentiation of SSCs and pre-BCSPs to BCSPs, resulting in defective commitment and differentiation of skeletal progenitors to osteoblasts.

To test whether Ecsit deletion in skeletal progenitors is a primary cause of impaired fracturing healing in Ecsit^Prx1 mice, Ecsit^fl/fl mice were crossed with Prx1-Cre^ERT2-EGFP mice (Ecsit^Prx1-ERT/GFP) to delete ECSIT expression in skeletal progenitors in an inducible manner. Fracture surgery was performed on the femurs of 6-week-old Ecsit^Prx1-ERT/GFP and Prx1-ERT/GFP control mice three days after tamoxifen injection (Supplemental Fig. 4a). One month later, tamoxifen-induced expression of Cre recombinase and deletion of Ecsit were validated by fluorescence microscopy and RT-PCR, respectively (Supplemental Fig. 4b, c). MicroCT analysis revealed that basal bone mass in control and Ecsit^Prx1-ERT/GFP femurs was comparable, as shown by equivalent trabecular bone mass and cortical bone thickness of non-fractured femurs (Supplemental Fig. 4d, e). Unlike Ecsit^Prx1 mice, tamoxifen-treated Ecsit^Prx1-ERT/GFP mice displayed normal periosteal and callus formation and fracture unionization in the fractured sites (Supplemental Fig. 4b, f, g). These results demonstrate that inducible deletion of Ecsit in Prx1+ skeletal

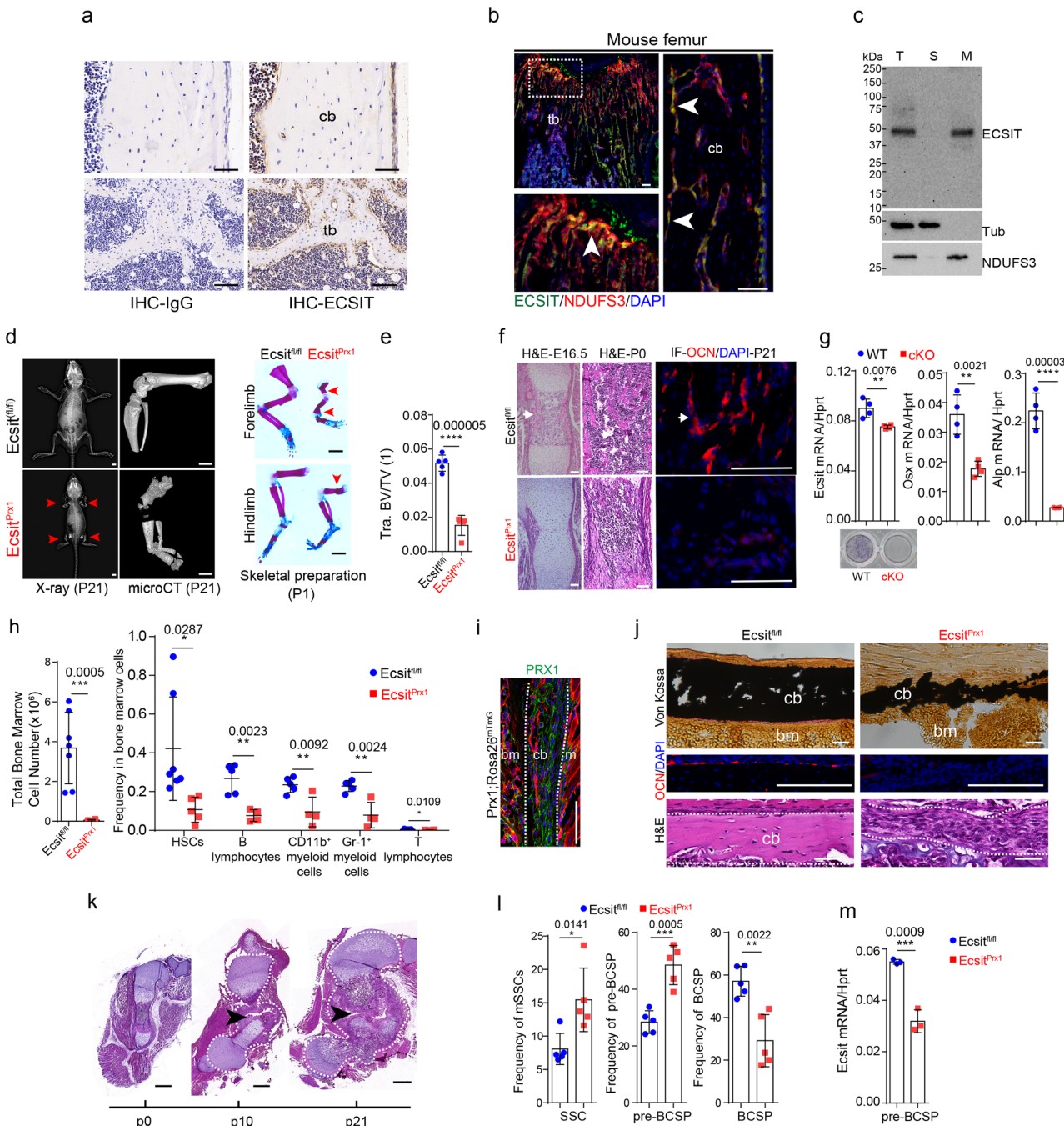

**Fig. 1 | *Ecsit^Prx1* mice show defects in skeletogenesis, fracture healing, and the bone marrow niche. a** Immunostaining of ECSIT in 2-month-old wild-type femurs. tb: trabecular bone, cb: cortical bone. **b** Immunofluorescence showing the localization of ECSIT in P21 mouse femurs. Arrows indicate co-localization of ECSIT and NDUFS3. **c** Immunoblot analysis showing mitochondrial fractionation of ECSIT in human BMSCs. **d, e** X-radiography (**d, left**) and microCT showing P21 whole body and hindlimbs (**d, middle**) and quantification of femoral bone mass (**e**, *n* = 5). Alizarin red/alcian blue staining of P1 neonates (**d, right**). Arrows indicate spontaneous fractures. Tra. BV/TV: Trabecular bone volume/total volume. **f** H&E staining of E16.5 and P0 femurs (**left, middle**). Immunofluorescence showing osteocalcin (OCN) expression in the femurs (**right**). Arrows indicate femoral trabecular bone. **g** Skeletal progenitors isolated by FACS from P0 *Ecsit;^Prx1Rosa26^mTmG* (cKO) and *Prx1;Rosa26^mTmG* (WT) limbs using GFP⁺CD45⁻Ter119⁻Tie2⁻ markers. Osteogenic gene expression (*n* = 4) and alkaline phosphatase activity (ALP, *n* = 3) were assessed for osteogenic differentiation. **h** Flow cytometry analysis showing frequency of the indicated bone marrow cell populations at P10 (**right**, HSCs (*Ecsit^fl/fl^*, *n* = 7; *Ecsit^Prx1^*, *n* = 5); other subpopulations, *n* = 4–5) and total number of the isolated cells (**left**, *Ecsit^fl/fl^*, *n* = 7; *Ecsit^Prx1^*, *n* = 6). Supplemental Fig. 2 demonstrates the gating strategy for this analysis. **i** GFP-expressing PRX1-lineage cells residing in the diaphyseal periosteum of P1 *PRX1-cre;Rosa26^mTmG* mice. m: muscle, bm: bone marrow. **j** P21 femoral cortical bones were stained for Von Kossa (**top**), H&E (**bottom**) and immunofluorescence for OCN (**middle**). **k** H&E staining of *Ecsit^Prx1^* femurs at the age of P0, P10, and P21. **l, m** Flow cytometry analysis showing frequency of SSCs, pre-BCSP, and BCSP in skeletal cells (**l**, *n* = 5). RT-PCR shows expression of *Ecsit* in pre-BCSP (**m**, *n* = 3). Supplemental Fig. 3 demonstrates the gating strategy for this analysis. A two-tailed unpaired Student's *t*-test for comparing two groups (**e, g, h, l, m**; error bars, data represent mean ± SD). Data are representative of three independent experiments. Scale bars = **a**, 100 μm; **b**, 50 μm; **d (left, middle)**, 2 mm; **d (right)**, 1 mm; **f**, 60 μm; **i**, 75 μm; **j**, 40 μm; **k**, 500 μm.

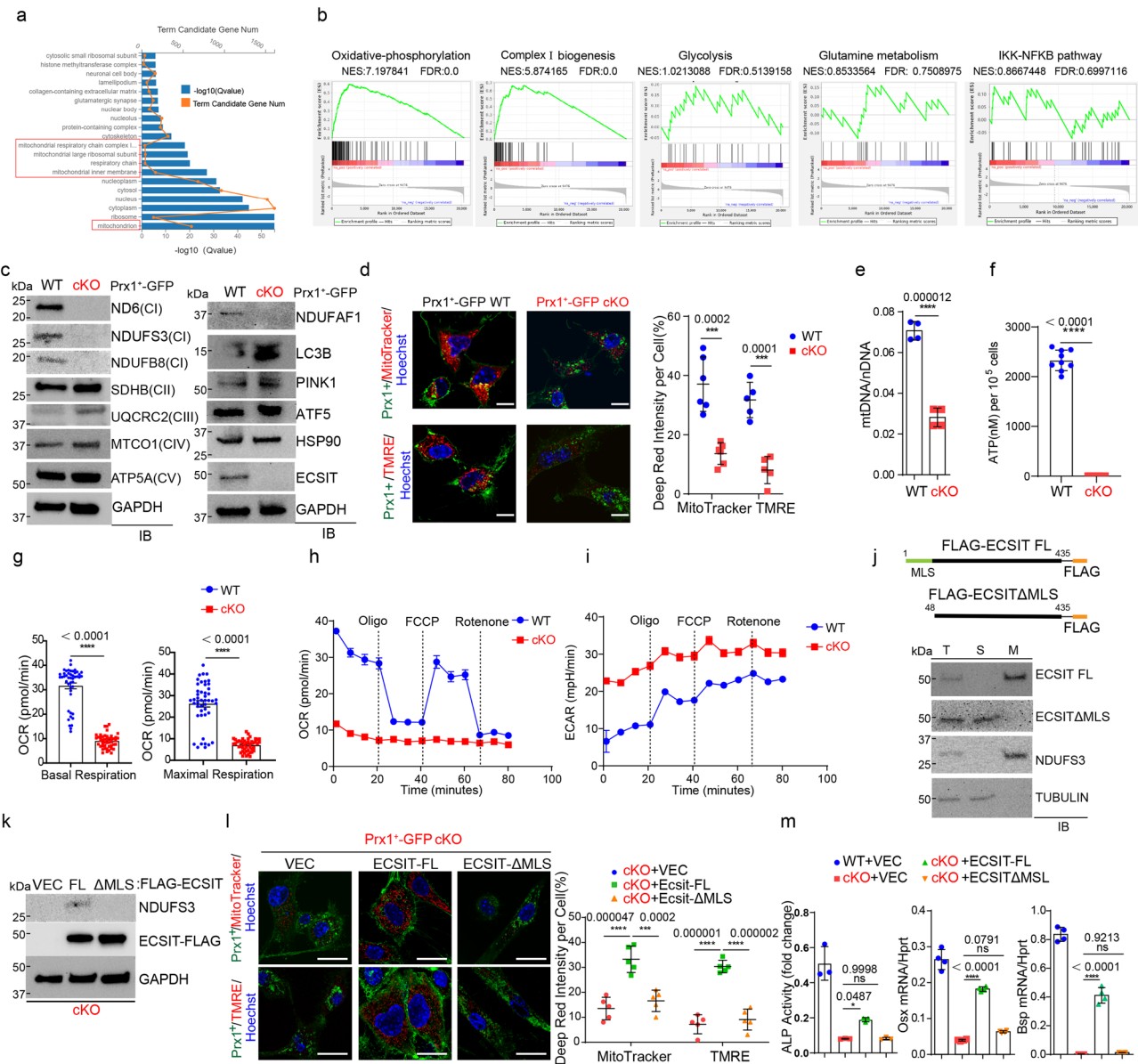

**Fig. 2 | *Ecsit*-deficiency in skeletal progenitors impairs mitochondrial OXPHOS.**
**a**, **b** Gene ontology analysis showing biological processes with gene enrichment in cKO vs WT skeletal progenitors (**a**). Gene set enrichment analysis (GSEA) showing the enrichment of genes involved in OXPHOS, CI biogenesis, glycolysis, glutamine metabolism, and IKK-NFκB signaling (**b**). **c** Immunoblot analysis showing protein levels in cKO and WT skeletal progenitors. GAPDH was used for a loading control. **d** MitoTracker and TMRE staining (red) of GFP-expressing cKO and WT skeletal progenitors (green). DAPI was used for nuclear staining (**left**). The signal intensity of the stained cells was quantified using ImageJ software (**right**). (MitoTracker (WT), $n = 6$; MitoTracker (cKO), TMRE (WT, cKO), $n = 5$. **e** RT-PCR analysis showing the ratio of mitochondrial DNA (mtDNA, Ndufv1) to nuclear DNA (nDNA,18s) in cKO and WT skeletal progenitors ($n = 4$). **f** Intracellular ATP levels in cKO and WT skeletal progenitors were measured by colorimetric analysis ($n = 9$). **g–i** Seahorse assay showing a real time oxygen consumption rate (OCR, **g–h**) and extracellular acid-ification rate (ECAR, **i**) of cKO and WT skeletal progenitors before and after the

addition of mitochondrial inhibitors ($n = 46$). **j** Diagram of constructs for ECSIT-full length and the mutant lacking mitochondrial localization sequence (MLS, 1-47 aa). A FLAG epitope tag was inserted into the C-terminus of ECSIT cDNA (**top**). Immunoblot analysis showing mitochondrial fractionation of FLAG-ECSIT proteins in cKO skeletal progenitors. (**bottom**). T total, S soluble, M mitochondrial fraction. **k** Immunoblot analysis showing expression of FLAG-ECSIT proteins in AAV-transduced cKO skeletal progenitors. **l**, **m** MitoTracker or TMRE staining of vector control or FLAG-ECSIT-expressing cKO and WT skeletal progenitors and quantifi-cation of deep red signal intensity (**l**). AAV-transduced cells were cultured under osteogenic conditions and ALP activity and osteogenic gene expression were assessed (**m**, $n = 4$). A two-tailed unpaired Student's *t*-test for comparing two groups (**d**, **e**, **f**, **g**) or ordinary one-way ANOVA with Dunnett's multiple comparisons test (**l**,**m**). (**d–i**; **l–m**; data represent mean ± SD). Data (**e**, **f**, **g**, **m**) are representative of three independent experiments. Scale bars = **d** (**left**), **l**, 10 μm.

progenitors in adult mice does not affect bone fracture healing and homeostasis. Thus, impaired fracture healing seen in *Ecsit^{Prx1}* mice may result from pre-existing skeletal phenotypes and/or alteration of ske-letal progenitor population during early skeletal development. How-ever, it cannot be fully excluded that inadequately complete deletion of *Ecsit* in this system contributes to the lack of a phenotype observed. Taken together, ECSIT-mediated mitochondrial functions play critical

roles in osteogenic commitment and differentiation of skeletal pro-genitors during early skeletal development.

## ECSIT controls mitochondrial OXPHOS in skeletal progenitors
Despite the original identification of ECSIT as a key regulator of NF-κB signaling downstream of pro-inflammatory receptors, Toll-like recep-tor (TLR) and interleukin (IL)-1β[38,39], transcriptome analysis showed an

enrichment of genes associated with metabolic pathways and mitochondrial function, not the NF-κB pathway in cKO *Prx1⁺* skeletal cells (Fig. 2a, b). Specifically, expression of the genes involved in mitochondrial CI biogenesis and OXPHOS was markedly downregulated in the absence of Ecsit while there was little to no altered expression in glucose and glutamine metabolism (Fig. 2b). This is consistent with previous reports demonstrating ECSIT as a key component of the MCIA complex that controls the stability of CI subunits[31–33]. To directly test the role of ECSIT in mitochondrial OXPHOS, the expression of mitochondrial CI, II, III, IV, and V subunit proteins in the OXPHOS pathway was examined in WT and cKO *Prx1⁺* skeletal progenitors, demonstrating that expression of core CI subunit proteins, including ND6, NDUFB8 and NDUFS3, was ablated in the absence of Ecsit while *Ecsit* deletion minimally affected the expression of proteins in other mitochondrial complexes (Fig. 2c, left). Similarly, *Ecsit*-deficiency significantly reduced expression of NDUFAF1, a key component of the MCIA complex[32,33] (Fig. 2c, right), suggesting the importance of ECSIT in stabilizing the components of CI and MICA complexes in *Prx1⁺* skeletal progenitors. Accompanying our transcriptome analysis that show little to no enrichment of genes associated with mitochondrial biogenesis (Fig. 2b), mRNA and protein levels of Atf5[40,41] and Hsp90[42,43] were largely intact in the absence of Ecsit (Fig. 2c, Supplemental Fig. 5a). Of note, expression of Lc3b and Pink1[44], key regulators of mitophagy, was markedly increased in cKO *Prx1⁺* skeletal progenitors (Fig. 2c), suggesting enhanced mitophagy activity. These results suggest that *Ecsit* deletion in *Prx1⁺* skeletal progenitors destabilizes the components of the MCIA and CI complexes while enhancing mitophagy activity. Accordingly, cKO *Prx1⁺* skeletal progenitors displayed a significant decrease in mitochondrial numbers and transmembrane potential (Fig. 2d), in the ratio of mitochondrial to nuclear DNA levels (Fig. 2e), and in ATP production (Fig. 2f). Along with a decrease in basal and maximal mitochondrial oxygen consumption (Fig. 2g), cKO *Prx1⁺* skeletal progenitors failed to respond to mitochondrial respiratory modulators, demonstrating the nearly complete absence of oxygen consumption rates (OCR, Fig. 2h). Of note, the glycolytic proton efflux rate was markedly upregulated in these cells, as shown by greater extracellular acidification rate (ECAR, Fig. 2i). However, enhanced glycolysis was incapable of compensating for the reduced ATP production by the damaged mitochondrial OXPHOS. In contrast to the involvement of glucose metabolism in RUNX2-mediated osteogenesis[45], expression and transcription activity of RUNX2 were largely intact in cKO *Prx1⁺* skeletal progenitors (Supplemental Fig. 5b, c), suggesting that ECSIT-mediated mitochondrial oxidative metabolism is dispensable for RUNX2-mediated osteogenesis.

Since ECSIT is localized on the membrane of mitochondria via an N-terminal mitochondria localization sequence (MLS, 1–48 aa)[32,46], to investigate mechanistic actions of ECSIT in regulating mitochondrial OXPHOS, cKO *Prx1⁺* skeletal progenitors were reconstituted with FLAG-tagged ECSIT full length (ECSIT-FL) or a MLS-deletion mutant (ECSIT-ΔMLS) that failed to localize to mitochondria via rAAV-mediated delivery (Fig. 2j). Enforced expression of ECSIT-FL, not the ECSIT-ΔMLS mutant, reversed alterations in expression of the CI subunit NDUFS3, mitochondrial numbers and transmembrane activity, and osteogenic potential in cKO *Prx1⁺* skeletal progenitors (Fig. 2k–m). Thus, mitochondrial ECSIT regulates CI assembly and mitochondrial OXPHOS in *Prx1⁺* skeletal progenitors, which in turn determines the osteogenic commitment and differentiation of skeletal progenitors.

To confirm these findings in vivo, a bone-targeting rAAV was used to express FLAG-tagged ECSIT-FL and ECSIT-ΔMLS proteins in *Ecsit^Prx1* mice (Fig. 3a). Given our previous report showing high transduction efficiency of the AAV9 serotype in osteoblast-lineage cells in vivo[47], systemic delivery of AAV9 vector expressing mCherry to P1 *Prx1;Rosa^{mT/mG}* neonates was effective in transducing GFP-expressing *Prx1⁺* osteoblast-lineage cells, including skeletal progenitors, osteoprogenitors, and mature osteoblasts, in both trabecular and cortical

compartments of long bones in addition to lung, heart, liver, kidney, and skeletal muscle (Fig. 3b, c). AAV-mediated expression of FLAG-tagged ECSIT-FL and ECSIT-ΔMLS proteins in the femur and liver was confirmed (Fig. 3f, Supplemental Fig. 6). Remarkably, enforced expression of ECSIT-FL, but not a vector control or ECSIT-ΔMLS, markedly improved the survival rates of *Ecsit^Prx1* mice (Fig. 3d). The osteogenic activity of *Ecsit^Prx1* skeletal progenitors was recovered by ECSIT-FL expression (Fig. 3f–h), leading to recovery of bone formation and regeneration activity at multiple anatomic compartments. This led to an almost complete rescue of skeletal deformities in *Ecsit^Prx1* forelimbs and hindlimbs (Fig. 3e) and abnormal bone marrow compartment (Fig. 3f, left) and impaired fracture healing and periosteal bone formation of *Ecsit^Prx1* long bones (Fig. 3g, h). However, none of these phenotypes were improved by AAV-mediated expression of ECSIT-ΔMLS (Fig. 3d–h). As expression of ECSIT-FL, not ECSIT-ΔMLS, restored OCN⁺ mature osteoblasts and osteoblast-supporting CD31⁺EDMC⁺ type H endothelium[48] in the fracture callus of *Ecsit^Prx1* long bones, fracture healing progressed normally in these mice (Fig. 3g, h). Restoring ECSIT levels in *Prx1⁺* osteoblast-lineage cells, including skeletal progenitors, osteoprogenitors, and mature osteoblasts, reverses skeletal defects and retrieves bone regeneration during fracture in *Ecsit^Prx1* mice. Thus, ECSIT-mediated regulation of mitochondrial OXPHOS in skeletal progenitors is crucial for bone accrual, fracture repair, and bone marrow niche maintenance during early skeletal development.

## ECSIT is dispensable of mitochondrial OXPHOS in committed skeletal cells

Since ECSIT is highly expressed in osterix (OSX)⁺ committed osteoprogenitors and dentin matrix acidic phosphoprotein 1 (DMP1)⁺ mature osteoblasts and osteocytes (Figs. 1a and 4a), the role of ECSIT in these cells were examined using conditional deletion with OSX-cre (*Ecsit^Osx*) or DMP1-cre (*Ecsit^Dmp1*) mice. mRNA and protein levels of *Ecsit* in these mice were examined by RT-PCR, immunoblot, and immunohistochemistry analyses, demonstrating lack of Ecsit expression (Fig. 4b, d, g). Remarkably, unlike *Ecsit^Prx1* mice, little to no gross skeletal phenotypes were seen in two-month-old *Ecsit^Osx* and *Ecsit^Dmp1* mice. Bone morphology, trabecular bone mass, and cortical bone thickness were all normal in the long bones of these mice (Fig. 4c, e, Supplemental Fig. 7). In addition, osteogenic differentiation and bone fracture healing were largely intact in the absence of Ecsit (Fig. 4f, Supplemental Fig. 8). These results suggest that ECSIT function is dispensable for the development of committed osteoblast-lineage cells, including *Osx⁺* osteoprogenitors and *Dmp1⁺* mature osteoblasts and osteocytes.

To gain insights on this mechanism, mitochondrial OXPHOS was examined in *Ecsit^Osx* osteoprogenitors and *Ecsit^Dmp1* mature osteoblasts. Unlike *Ecsit^prx1* skeletal progenitors, expression of the CI protein NDUFS3, ND6, and NDUFB8, and the MCIA subunit NDUFAF1 (Fig. 4g), the ratio of mitochondrial DNA to nuclear DNA (Fig. 4h), mitochondrial numbers and transmembrane potential (Fig. 4i) were all intact in these cells, suggesting that ECSIT function is dispensable for mitochondrial OXPHOS in *Osx⁺* committed osteoprogenitors and *Dmp1⁺* mature osteoblasts and osteocytes. Notably, while a similar decrease in ECSIT expression was observed among *Ecsit^Prx1* skeletal progenitors, *Ecsit^Osx* osteoprogenitors, and *Ecsit^Dmp1* osteoblasts, only *Ecsit^Prx1* cells showed a significant decrease in NDUFS3 expression (Supplemental Fig. 9). However, it cannot be fully excluded that inadequately deletion of *Ecsit* in osteoblasts may contribute to the lack of an observed phenotype. Thus, the role of ECSIT in controlling mitochondrial oxidative metabolism is context- and tissue-dependent.

## *Ecsit* deficiency in skeletal progenitors induces muscle atrophy

Accompanying the onset spontaneous fractures in *Ecsit^Prx1* mice is an onset of myopathy and muscle atrophy. Specifically, the size and

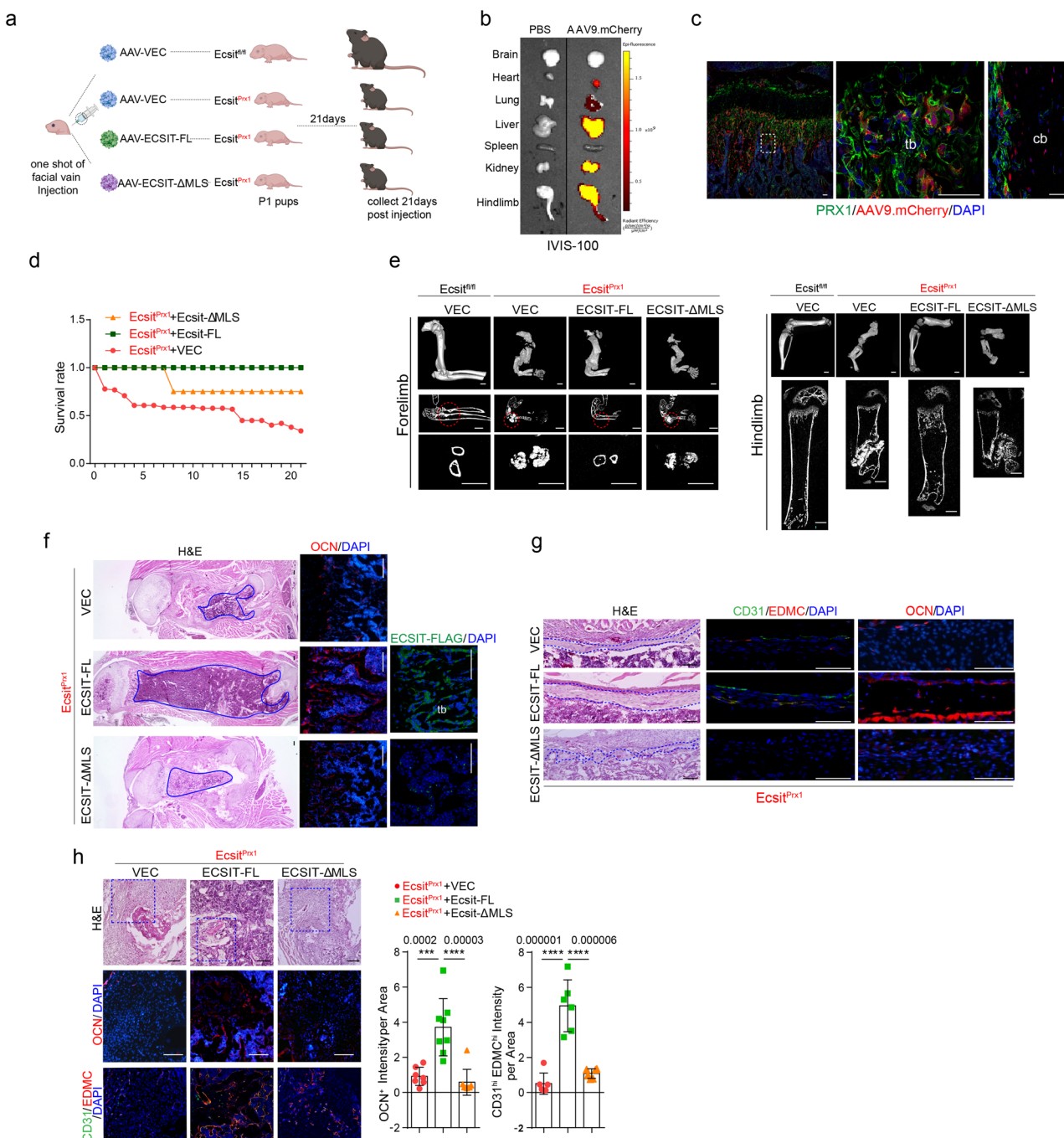

**Fig. 3 | AAV-mediated expression of ECSIT reverses *Ecsit^Prx1* skeletal phenotypes. a** Diagram summarizing the study and treatment methods. A single dose of $2 \times 10^{11}$ genome copies (GCs) of rAAV9 vectors carrying control vector or FLAG-ECSIT constructs was injected into P1 *Ecsit^Prx1* neonates via the facial vein and musculoskeletal phenotypes were assessed 21 days post-injection (created with biorender.com). **b** Single dose of $2 \times 10^{11}$ GCs of rAAV9 carrying mCherry was injected into P0 *Prx1;Rosa26^mTmG* neonates via facial vein and mCherry expression was monitored by IVIS-100 optical imaging 21 days post-injection. **c** rAAV9 vector carrying mCherry was injected into *Prx1;Rosa26^mTmG* neonates and 21 days later, mCherry expression was assessed by fluorescence microscopy of cryo-sectioned femurs. **d** Survival rate of AAV-treated *Ecsit^Prx1* mice up to 21 days post-injection ($n = 6$). **e** MicroCT analysis of the forelimbs (**left**) and hindlimbs (**right**) of P21 AAV-treated *Ecsit^fl/fl* and *Ecsit^Prx1* mice ($n = 5-8$). **f** Longitudinal sections of P21 AAV-treated *Ecsit^fl/fl* and *Ecsit^Prx1* femurs were stained for H&E (**left**) or immunostained for OCN (**middle**) or FLAG-ECSIT (**right**). Blue lines indicate the bone marrow area. **g** H&E staining (**left**) or immunofluorescence for CD31, EDMC, or OCN (**middle, right**) of diaphyseal cortical bones of P21 AAV-treated *Ecsit^fl/fl* and *Ecsit^Prx1* femurs. Blue lines indicate periosteum. **h** H&E staining (**top**) or immunofluorescence for CD31, EDMC (VEC, $n = 7$; ECSIT-FL, $n = 8$, ECSIT-MLS, $n = 6$), or OCN ($n = 6$, **middle, bottom**) of fracture sites of P21 AAV-treated *Ecsit^fl/fl* and *Ecsit^Prx1* femurs. Immunofluorescence intensity was quantified using ImageJ software. An ordinary one-way ANOVA with Dunnett's multiple comparisons test (**h**; error bars, data represent mean ± SD). Data are representative of three independent experiments. Scale bars = **c**, 50 μm; **e**, 1 mm; **f**, **h**, 100 μm; **g**, 75 μm.

weight of skeletal muscles, including tibialis anterior (TA) and gastrocnemius (GA) muscles, were dramatically decreased (Fig. 5a, b, Supplemental Fig. 10a). While muscle structure of *Ecsit^Prx1* neonates is grossly normal after birth (Fig. 5c, Supplemental Fig. 10b), *Ecsit^Prx1* mice

gradually developed severe degeneration of skeletal muscle, as evident by significant decrease in mature myofibers, extracellular matrix in the interstitial space, and the expression of mature myoblast genes (Fig. 5d, e, Supplemental Fig. 10c). Despite a substantial decrease in

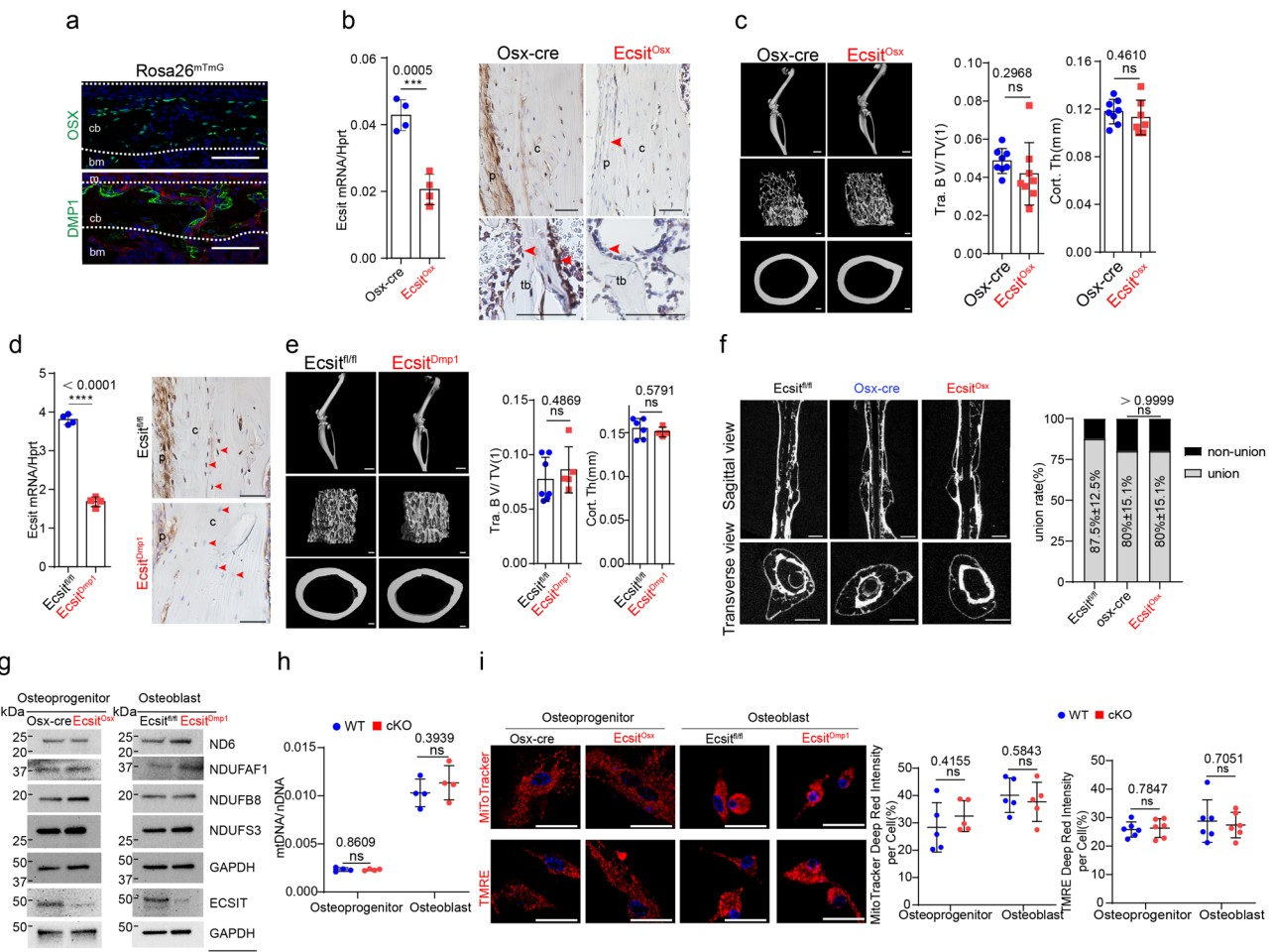

**Fig. 4 | *Ecsit^Osx* and *Ecsit^Dmp1* mice show no skeletal phenotype and intact mitochondrial function. a** GFP-expressing Osterix (OSX)-, DMP1-lineage cells residing in the diaphyseal periosteum of P1 *Osx-EGFP* and *DMP1-cre;Rosa26^mTmG* mice, respectively, were monitored by fluorescence microscopy. m: muscle, bm: bone marrow. **b, d** mRNA levels of *Ecsit* in tibia was examined by RT-PCR (*n* = 4, **left**). Immunohistochemistry showing ECSIT expression in 2-month-old *Ecsit^Osx*, *Ecsit^Dmp1*, and littermate control femurs (**right**). p: periosteum, tb: trabecular bone, c: cortical bone. Red arrows indicate *Osx^+* or *Dmp1^+* lineage osteoblasts or osteoctyes. **c, e** MicroCT analysis showing 3D-reconstruction images of hindlimbs, femoral trabecular and cortical bones of 2-month-old *Ecsit^Osx*, *Ecsit^Dmp1* and littermate control mice (**left**), and the relative quantification of trabecular bone mass (Osx-cre, *Ecsit^Osx*, *n* = 8; *Ecsit^fl/fl*, *n* = 7; *Ecsit^Dmp1*, *n* = 5) and cortical bone thickness (Osx-cre, *n* = 8; *Ecsit^Osx*, *n* = 7; *Ecsit^fl/fl*, *n* = 6; *Ecsit^Dmp1*, *n* = 4, **right**). Osx-cre limbs were used as controls for *Ecsit^Osx* mice. **f** MicroCT analysis showing sagittal and transverse views

of fracture sites of 2-month-old *Osx-cre*, *Ecsit^Osx*, and *Ecsit^fl/fl* femurs 10 weeks after the surgery, and the relative quantification of fracture union rates (*n* = 5).
**g–i** Osteoprogenitors and osteoblasts were isolated from *Osx-cre* and *Ecsit^Osx* calvaria and *Ecsit^fl/fl* and *Ecsit^Dmp1* long bones, respectively. Immunoblot analysis showing protein levels in the isolated cells. GAPDH was used as a loading control (**g**). A ratio of mtDNA (Ndufv1) to nDNA (18s) was assessed by RT-PCR (*n* = 4, **h**). Fluorescence microscopy shows MitoTracker- and TMRE-stained cells (**i, left**) and relative quantification of MitoTracker- (*n* = 5, **i, middle**) and TMRE-red signal intensity (*n* = 5, **i, right**). A two-tailed unpaired Student's *t*-test for comparing two groups (**b, c, d, e, h, i**; error bars, data represent mean ± SD) or a Fisher's exact test (**f**, *p* > 0.9999) was applied. Data are representative of three independent experiments. Scale bars = **a**, 75 μm; **b, d**, 100 μm; **c, e** (**left, top**) 2 mm; **c, e** (**left, middle and bottom**), 100 μm; **f**, 1 mm; **i** (**left**), 25 μm.

total myocyte numbers in *Ecsit^Prx1* mice, the frequency of skeletal muscle-resident satellite cells[49] (β-integrin^+CXCR4^+Sca1^-CD45^-CD11b^-TER119^-) and mesenchymal progenitors[50] (Sca1^+integrin α7^-CD45^-CD31^-) was dramatically increased in *Ecsit^Prx1* muscle (Fig. 5f, Supplemental Fig. 11), indicating a defect in myoblast differentiation, but not the maintenance of muscle-resident stem cells. Of note, this phenotype does not result from the intrinsic effects of *Ecsit* deficiency in muscle cells, as shown by intact expression of *Ecsit* mRNA and protein in *Ecsit^Prx1* muscle (Fig. 5d, e). This is accompanied by fluorescence microscopy showing little to no Prx1-cre driven reporter activity in skeletal muscle cells of *Prx1-cre;Rosa26^mT/mG* mice (Fig. 5g). To directly test this, mice with targeted deletion of *Ecsit* in myoblast precursors (*Ecsit^Myf5*) were generated, demonstrating little to no gross phenotypes in the skeletal muscle, bone, and bone marrow (Fig. 5h). Likewise, the expression of mature myoblast genes in the skeletal muscle of these

mice was also normal (Fig. 5i). Additionally, primary human myoblast progenitors (17Ubic)[51] with *Ecsit* knockdown were cultured under myogenic conditions, demonstrating no effect of *Ecsit*-deficiency on myoblast differentiation (Fig. 5j). As seen in *Osx^+* committed osteoprogenitors and *Dmp1^+* mature osteoblasts (Fig. 4g), deletion of *Ecsit* in myoblast progenitors did not affect expression of CI protein NDUFS3, the ratio of mitochondrial DNA to nuclear DNA, mitochondrial numbers and transmembrane potential while these cells showed a mild reduction in the MCIA subunit Ndufaf1 (Fig. 5k, l). These results suggest that ECSIT function is dispensable for mitochondrial OXPHOS for ATP production during myoblast differentiation. Thus, *Ecsit* deletion in *Prx1^+* skeletal progenitors, not muscle cells, is responsible for muscle atrophy in *Ecsit^Prx1* mice.

To determine how skeletal fracture could lead to muscle atrophy, the migration of GFP-expressing *Ecsit^Prx1* skeletal progenitors into

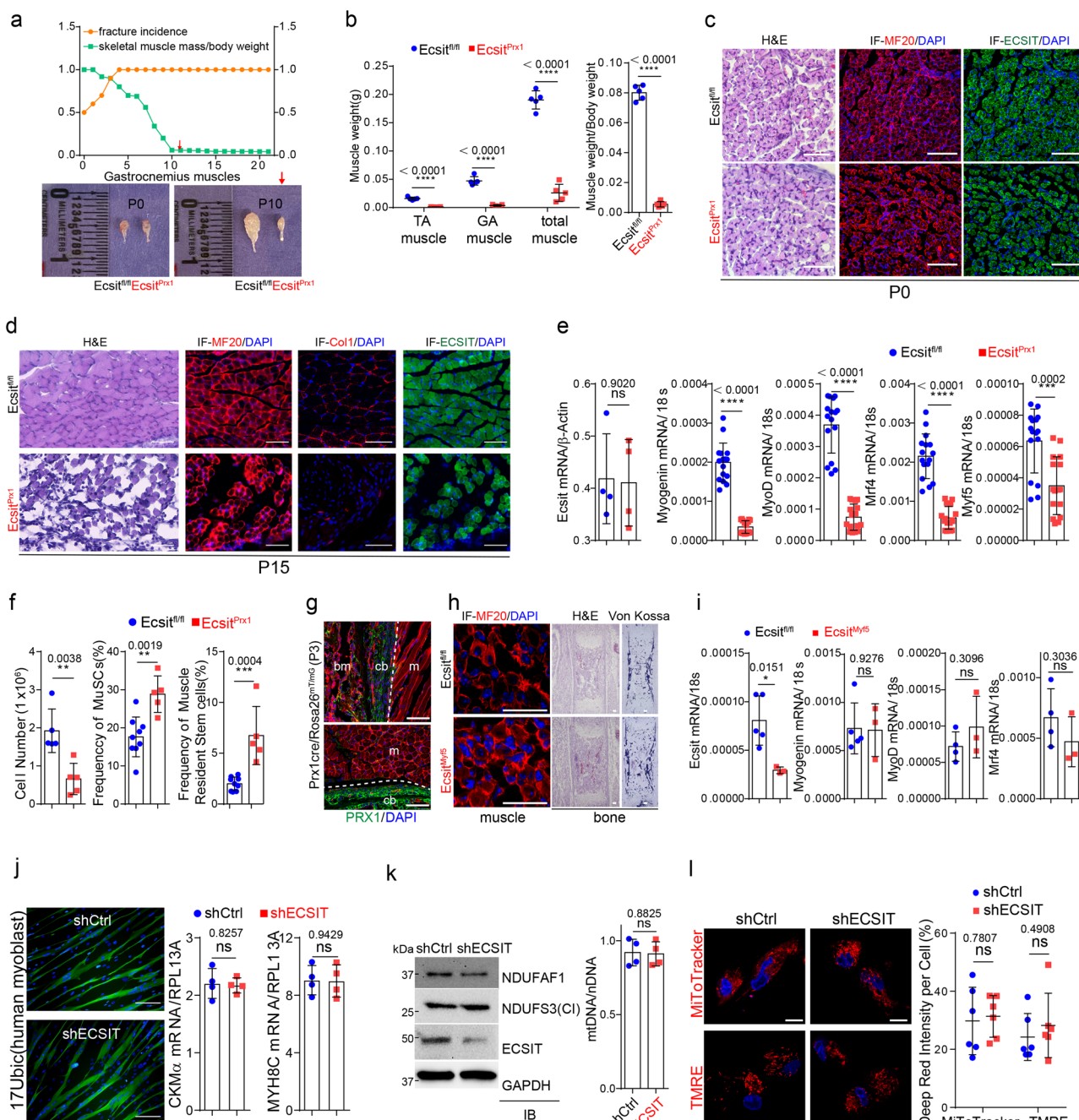

**Fig. 5 | *Ecsit^Prx1* mice display skeletal muscle atrophy. a** Graph shows the kinetics of fracture incidence and skeletal muscle weight of *Ecsit^Prx1* and *Ecsit^fl/fl* mice (*n* = 5, **left**). Representative macroscopic images of the gastrocnemius (GA) muscle at the age of P0 and P10 (**right**). **b** Quantification of skeletal muscle weight (**left**) and a ratio of skeletal muscle weight to body weight (**right**) are displayed (*n* = 5). **c, d** P0 (**c**) or P15 (**d**) *Ecsit^Prx1* and *Ecsit^fl/fl* tibialis anterior (TA) muscles were stained for H&E or immunostained for MF20, COL1α1, or ECSIT. **e** mRNA levels of myogenic genes in P10 *Ecsit^Prx1* and *Ecsit^fl/fl* TA muscles (*n* = 16). **f** Total cell number (*n* = 5) and flow cytometry analysis showing the frequency of muscle satellite cells (MuSCs) and skeletal muscle resident stem/progenitor cells (*Ecsit^fl/fl*, *n* = 9, *Ecsit^Prx1*, *n* = 5) in P10 *Ecsit^Prx1* and *Ecsit^fl/fl* GA muscles. Supplemental Fig. 11 demonstrates the gating strategy for this analysis. **g** GFP-expressing Prx1⁺ skeletal cells in P3 *Prx1-cre;Rosa26^mTmG* GA muscle. Red: Prx1⁻ bone marrow and muscle. **h, i** Tibialis anterior muscles

or femurs of E21 *Ecsit^Myf5* and *Ecsit^fl/fl* embryos were stained for H&E, Von Kossa, or immunostained for MF20 (**h**). mRNA levels of *Ecsit* and myoblast genes in the skeletal muscle (**i**, *Ecsit^fl/fl*, *n* = 4−5; *Ecsit^Myf5*, *n* = 3). **j** Human 17Ubic myoblasts expressing control (shCtrl) or *ECSIT* shRNA (shECSIT) were cultured under myogenic conditions. 6 days later, expression of MF20 (green) and myogenic genes was assessed (*n* = 4). **k, l** Protein levels of NDUFAF1, ECSIT and NDUFS3 (**k, left**) or the ratio of mtDNA (Ndufv1) to nDNA (18s, **k, right**) in shCtrl or shECSIT-expressing 17Ubic cells. Fluorescence microscopy shows MitoTracker- and TMRE-stained cells and relative quantification of deep red signal intensity (*n* = 6, **l**). A two-tailed unpaired Student's *t*-test for comparing two groups (**b, e, f, i, j, k, l**; error bars, data represent mean ± SD). Data are representative of three independent experiments. Scale bars = **c, d**, 60 μm; **g**, 75 μm; **h**, 30 μm; **j**, 100 μm; **l**, 10 μm.

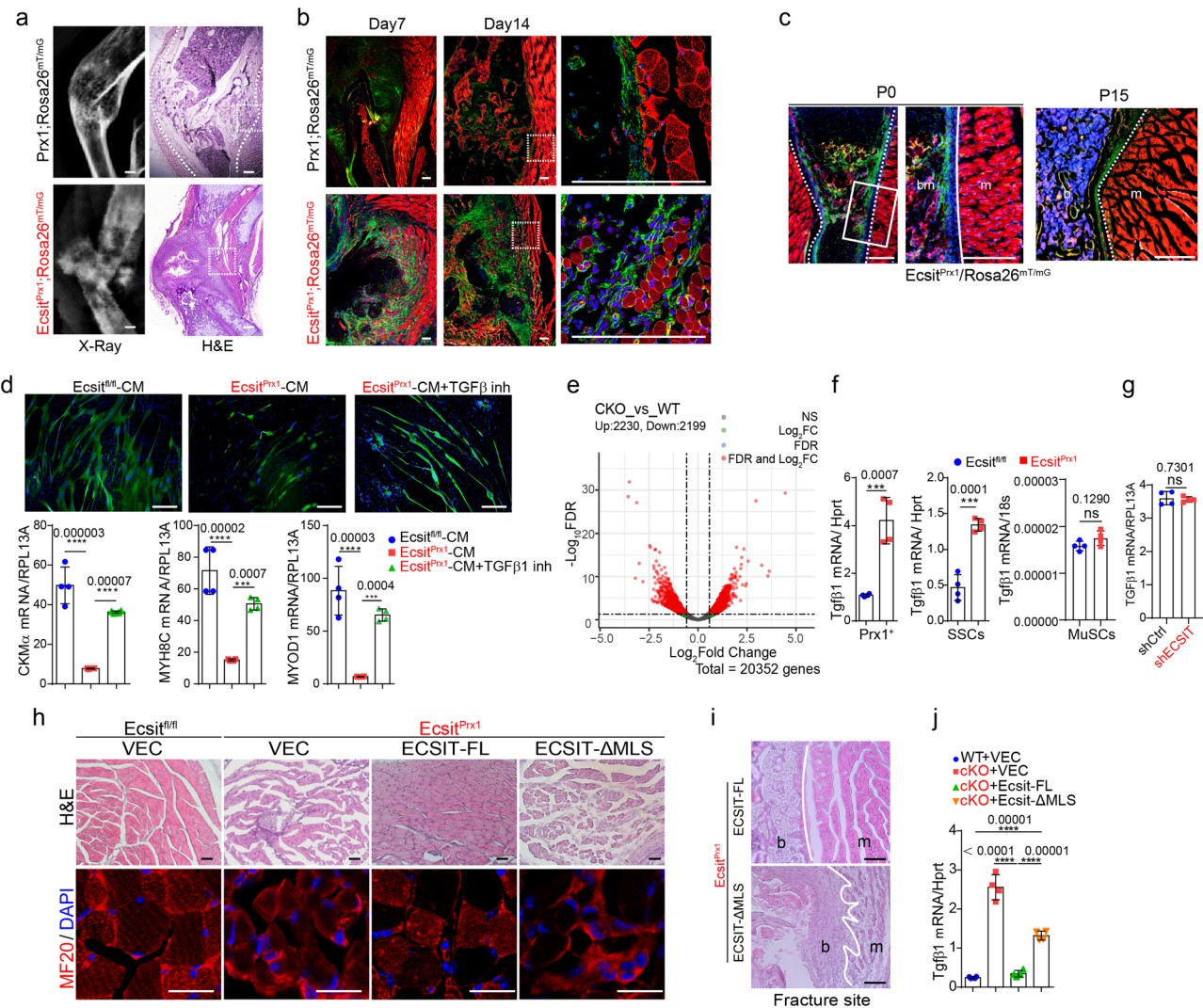

**Fig. 6 | *Ecsit^{Prx1}* skeletal progenitors induce skeletal muscle degeneration.**
**a** X-radiography (**left**) and H&E staining (**right**) of P14 *Ecsit;^{Prx1}Rosa26^{mTmG}* tibias with spontaneous fractures and 6-week-old *Prx1;Rosa26^{mTmG}* tibias 14 days post-fracture surgery. **b** Fluorescence microscopy showing GFP-expressing *Prx1^+* skeletal cells in P7 or P14 *Ecsit;^{Prx1}Rosa26^{mTmG}* tibias with spontaneous fracture and 6-week-old *Prx1;Rosa26^{mTmG}* tibias 7- or 14-days post-fracture surgery. Red: *Prx1^-* tissues. **c** Non-fractured sites in P0 (**left**) or P15 (**right**) *Ecsit;^{Prx1}Rosa26^{mTmG}* femurs, demonstrating intact skeletal muscle structure. Green: *Prx1^+* skeletal cells. Red: Prx1^- bone marrow and muscle. **d** Human 17Ubic myoblasts were cultured under myogenic conditions in the presence of conditioned medium (CM) harvested from the culture of FACS-sorted *Ecsit;^{Prx1}Rosa26^{mTmG}* (cKO) and *Prx1;Rosa26^{mTmG}* (WT) skeletal progenitors. 5 days later, expression of MF20 (green) and myogenic genes was assessed (*n* = 4). Alternatively, cells were treated with the TGF-β1 inhibitor SB431542. **e** RNA sequencing was performed on FACS-sorted *Ecsit;^{Prx1}Rosa26^{mTmG}* (cKO) and *Prx1;Rosa26^{mTmG}* (WT) skeletal progenitors. A volcano plot shows the gene expression for

up/downregulated genes in cKO cells relative to WT cells. **f** mRNA levels of Tgf-β1 in FACS-sorted skeletal progenitors (**left**), SSCs (**middle**), or skeletal muscle satellite cells (MuSCs, **right**) (*n* = 4). **g** mRNA levels of Tgfβ1 in 17Ubic myoblast progenitors expressing control (shCtrl) or *ECSIT* shRNA (shECSIT) (*n* = 4). **h, i** AAV-treated *Ecsit^{fl/fl}* and *Ecsit^{Prx1}* TA muscles (P21) were stained for H&E (**h, top**) or immunostained for MF20 (**h, bottom**). H&E staining demonstrate intact cortical bone, periosteum, and skeletal muscle in *Ecsit^{Prx1}* mice expressing FLAG-tagged ECSIT-FL, not ECSIT-ΔMLS (**i**). White lines indicate the barrier between bone and skeletal muscle. **j** mRNA levels of Tgf-β1 in vector control or FLAG-ECSIT-expressing cKO and WT skeletal progenitors (*n* = 4). A two-tailed unpaired Student's t-test for comparing two groups (**f**, **g**) or ordinary one-way ANOVA with Dunnett's multiple comparisons test (**d, j**) (**d, f, i**, data represent mean ± SD). Data are representative of three independent experiments. Scale bars = **a**, **c**, 200 μm; **b**, **d**, 100 μm; **h**, 75 μm; **i**, 20 μm.

skeletal muscle was monitored at fracture sites. Wild-type mice undergoing fracture did not experience muscle degeneration, showing normal callus formation at 7- and 14-days post-fracture, and restriction of wild-type *Prx1*-lineage skeletal cells to the callus and not adjacent skeletal muscle (Fig. 6a). On the other hand, *Ecsit^{Prx1}* mice showed abnormal callus development and fibrotic tissue at fracture sites and failed to form periosteum, where *Ecsit^{Prx1}* skeletal progenitors migrated into the skeletal muscle and induced degeneration of myofibers (Fig. 6b). In non-fractured long bone showing normal muscle structure, *Ecsit^{Prx1}* skeletal progenitors were only detected inside the bone (Fig. 6c). Thus, impaired fracture healing and periosteal formation

allow migration of *Ecsit^{Prx1}* skeletal progenitors to adjacent skeletal muscle, resulting in muscle atrophy. To directly test this hypothesis, conditioned media (CM) harvested from WT or cKO *Prx1^+* skeletal progenitors were added to cultures of human myoblast progenitors (17Ubic). These studies demonstrated a significant reduction in myoblast differentiation in the presence of cKO-CM compared to WT-CM (Fig. 6d). Transcriptome and RT-PCR analyses of FACS-sorted WT and cKO *Prx1^+* skeletal progenitors and SSCs revealed elevated levels of the soluble myogenic suppressor TGF-β1[52,53] in cKO cells (Fig. 6e, f). Since TGF-β1 expression was unchanged in satellite cells (MuSCs) isolated from the skeletal muscle of *Ecsit^{Prx1}* mice and *Ecsit*-deficient myoblast

progenitors (Fig. 6f, g), cKO *Prx1*⁺ skeletal progenitors and SSCs, not MuSCs and myoblast progenitors, are likely to be major contributors of TGF-β1 expression in skeletal muscle. Accordingly, the TGF-β1 inhibitor SB-431542 almost completely reversed the ability of cKO-CM to suppress myoblast differentiation (Fig. 6d). Since the TGF-β pathway plays a role in the pathogenesis of skeletal muscle myopathies and its inhibition is considered for the treatment of skeletal muscle fibrosis[54], TGF-β1 is likely to be an important factor mediating the myoblast suppressive activity of *Ecsit^{Prx1}* skeletal progenitors, though other pathways may contribute as well.

Intravenous administration of P1 *Ecsit^{Prx1}* neonates with bone-targeting rAAV9 expressing FLAG-tagged ECSIT-FL, not ECSIT-ΔMLS mutant, almost completely recovered degeneration of skeletal muscle (Fig. 6h). This was accompanied with little to no migration of *Ecsit^{Prx1}* skeletal progenitors expressing ECSIT-FL, not ECSIT-ΔMLS mutant, to the skeletal muscle (Fig. 6i). Finally, enforced expression of ECSIT-FL markedly decreased TGF-β transcription in FACS-sorted cKO *Prx1*⁺ skeletal progenitors while TGF-β expression was partially reversed by ECSIT-ΔMLS mutant (Fig. 6j), suggesting the presence of additional functional domain(s) within Ecsit regulating TGF-β expression. Thus, impaired OXPHOS in skeletal progenitors is likely to be responsible for muscle atrophy seen in *Ecsit^{Prx1}* mice. Taken together, ECSIT-mediated regulation of mitochondrial oxidative metabolism is a key determinant of the osteogenic commitment and differentiation of *Prx1*⁺ skeletal progenitors, governing musculoskeletal development and bone marrow niche functions.

## Discussion

Our study provides in vivo evidence showing the importance of mitochondrial oxidative metabolism in determining osteogenic commitment and differentiation of skeletal progenitors during early skeletal development. Despite emerging evidence suggesting an association of mitochondrial dysfunction and high oxidative stress in skeletal cells with skeletal disorders[17], only few genetic models with mitochondrial OXPHOS defects have been successfully generated due to either embryonic lethality or the absence of any relevant phenotypes. Herein, we generated a mouse model with CI deficiency by conditionally deleting the key CI assembly factor *Ecsit* in various musculoskeletal cells. Deletion of *Ecsit* in *Prx1*⁺ skeletal progenitors disrupts mitochondrial OXPHOS for ATP production while enhancing glucose metabolism as a compensatory mechanism. Lack of OXPHOS activity ablates the osteogenic potential of skeletal progenitors, resulting in defects in skeletal development, bone marrow niche maintenance, and bone fracture repair. Surprisingly, breakdown of this regulatory circuit leads to the infiltration of *Prx1*⁺ skeletal progenitors into adjacent muscle, which induces myopathy mediated by TGF-β1 (Supplemental Fig. 12). This unexpected finding indicates that OXPHOS defects or TGF-β1 may be implicated in paired skeletal fragility/myopathy disorders, and there may be potential therapeutic targets in this context. Intriguingly, despite the requirement of mitochondrial localization of Ecsit in regulating mitochondrial oxidative metabolism in *Prx1*⁺ skeletal progenitors, our findings that cytosolic Ecsit (Ecsit-ΔMLS) can partially improve survival of *Ecsit^{Prx1}* mice and TGF-β1 expression in *Ecsit^{Prx1}* skeletal progenitors suggest the presence of additional functional domain(s) within Ecsit regulating survival and TGF-β1 expression. Further studies will be necessary to identify these functional domains in *Prx1*⁺ skeletal progenitors.

Glucose and glutamine metabolism and mitochondrial OXPHOS have been known as major pathways of energy metabolism regulating the production of ATP in osteoblast-lineage cells[24]. In particular, glucose and glutamine metabolism are prominent metabolic features of committed osteoblast-lineage cells[55–58] and integral to collagen production and mineralizing activity by generating ATP and intermediate metabolites[59,60]. In contrast to our findings that ECSIT-mediated mitochondrial OXPHOS provides an energy source crucial to the osteogenic commitment and differentiation of *Prx1*⁺ skeletal progenitors during skeletal development, glutamine metabolism mainly contributed to biosynthesis and redox homeostasis, but not energy production, during bone remodeling. Of note, since amino-acid-transaminase-derived αKG is the critical downstream glutamine metabolite regulating proliferation of skeletal progenitors[29], cell proliferation rates were largely not altered in *Ecsi^{Prx1}* skeletal progenitors with normal glutamine metabolism. Intriguingly, despite extensive studies showing that glucose is a major nutrient for osteoblast development, mice lacking the glucose transporter Glut1 in *Prx1*⁺ skeletal progenitors displayed impaired cartilage development without any alteration in skeletal mineralization[28]. This suggests that glycolysis is critical for chondrogenesis, not osteogenesis, of *Prx1*⁺ skeletal progenitors during skeletal development. Notably, *Ecsit*-deficiency in *Prx1*⁺ skeletal progenitors enhanced glucose metabolism as a compensatory mechanism while cartilage development occurred normally in *Ecsi^{Prx1}* mice. Thus, our data implies that mitochondrial OXPHOS, not glucose or glutamine metabolism, is the major energy-driving force for osteogenesis in skeletal progenitors and that it is, therefore, critical for skeletal development and the maintenance of musculoskeletal integrity. We note that reduced mitochondrial mass in *Ecsi^{Prx1}* skeletal progenitors also potentially results in a broader set of cellular abnormalities than just decreased ATP synthesis, as many TCA cycle metabolites, such as acetyl-CoA and α-ketoglutaric acid, have direct roles in lipid synthesis, protein acetylation, histone demethylation, and other processes[29,61,62].

Since ECSIT function is dispensable for *Osx*⁺ committed osteoprogenitors and *Dmp1*⁺ mature osteoblasts and terminally differentiated osteocytes, glycolysis may offset this as glycolysis is the primary metabolic pathway for ATP production during the differentiation of committed osteoblast-lineage cells[59]. Further studies will be necessary to determine how ECSIT controls mitochondrial oxidative metabolism and osteogenic development in a context and tissue-dependent manner. Additional further studies are warranted to define the precise mechanisms by which mitochondrial OXPHOS impacts skeletal progenitors through other mitochondrial regulatory mechanisms, such as mitochondrial biogenesis, the degradation of damaged mitochondria via mitophagy, reactive oxygen species (ROS) generation, and mtDNA replication and/or transcription.

The unexpected finding that mitochondrial defects lead *Ecsit^{Prx1}* skeletal progenitors to infiltrate into muscle and mediate pathogenic TGF-β1 secretion, provides a new potential explanation for a number of clinical disorders that pair prominent or primary skeletal phenotypes with myopathy. For instance, Kearns Sayre Syndrome displays linked skeletal and myopathy phenotypes due to mitochondrial dysfunction, raising the possibility that the myopathy may be in part secondary to pathogenic infiltration of skeletal progenitors into adjacent muscle[20]. rAAVs have been considered the most promising viral vector for gene therapy[63], as they have a clinical track record of being well tolerated and have the potential to efficiently mediate genetic modifications that can persist for years after a single treatment. Remarkably, AAV-mediated expression of full length Ecsit almost completely restored impaired mitochondrial OXPHOS and osteogenic potentials of *Ecsit^{Prx1}* progenitors, preventing musculoskeletal deformities of *Ecsit^{Prx1}* mice. Thus, our findings provide proof-of-concept demonstration for gene therapy of developmental skeletal disorders. Collectively, this study reveals a previously unappreciated role of mitochondrial oxidative metabolism in determining osteogenesis of skeletal progenitors and demonstrates a new gene therapy approach for mitochondrial diseases in the musculoskeletal system.

## Methods
### Mice
Mice were housed in a constant environment under a standard mouse chow diet (up to 5 mice per cage); ambient temperature of $21 \pm 2 \,°C$,

circulating air, and constant humidity of $50 \pm 10\%$, in a 12 h light, 12 h dark cycle. Mice were monitored every three days for the amount of their food and water intake and signs for distress. For signs of severe distress, including general malaise, severe cachexia, or more than 20% loss of body weight, humane euthanasia was performed in consultation with veterinary staff. Mice were euthanized in a carbon dioxide chamber, followed by cervical dislocation. Neonates were euthanized by decapitation. *Ecsit*$^{fl/fl}$ mice were generated as previously described and maintained on a C57BL/6J background[46]. Cre deleter mice (C57BL/6J) that express Cre recombinase under the control of the prx1 promoter (Prx1-cre), the osterix promoter (Osx-cre), and the dmp1 promoter (Dmp1-cre) were purchased from The Jackson Laboratory. To label Cre-expressing cells, *Ecsit;*$^{fl/fl}$*Prx1-cre* mice were further crossed with *Rosa*$^{mT/mG}$ cre reporter mice (C57BL/6J)[64]. Mouse genotypes were determined by PCR on tail genomic DNA and primer sequences are available upon request. Control littermates were used and analyzed in all experiments. A phenotypic summary of *Ecsit*$^{Prx1}$ mice is described in Supplemental Table 1. All animals were used in accordance with the NIH Guide for the Care and Use of Laboratory Animals and were handled according to protocols approved by the University of Massachusetts Medical School Institutional Animal Care and Use Committee (IACUC).

## Generation of rAAV vectors

DNA sequences for ECSIT-FL and ECSIT-ΔMSL were synthesized as gBlocks, cloned into the intronic region of the pAAVsc-CB6-Egfp plasmid at the restriction enzyme sites (PstI and BglII)[65], and packaged into an AAV9 capsid. rAAV production was performed by transient transfection of HEK293 cells, purified by CsCl sedimentation, titered by droplet digital PCR (ddPCR) on a QX200 ddPCR system (Bio-Rad) using the Egfp prime/probe set as previously described[47,65]. The sequences of gBlocks and oligonucleotides for ddPCR and are listed in Supplemental Table 2.

## MicroCT, radiography, and skeletal preparation

MicroCT was used for qualitative and quantitative assessment of trabecular and cortical bone microarchitecture and performed by an investigator blinded to the genotypes of the animals under analysis. Femurs excised from the indicated mice were scanned using a microCT 35 (Scanco Medical) with a spatial resolution of 7 μm. For trabecular bone analysis of the distal femur, an upper 2.1 mm region beginning 280 μm proximal to the growth plate was contoured. For cortical bone analysis of the femur and tibia, a midshaft region of 0.6 mm in length was used. MicroCT scans of forelimbs and hindlimbs were performed using isotropic voxel sizes of 12 μm. 3D reconstruction images were obtained from contoured 2D images by methods based on distance transformation of the binarized images. All images presented are representative of the respective genotypes ($n = 5$–8).

X-radiographic images of whole body and hindlimbs were taken by the Trident Specimen Radiography System (Hologic, Inc.).

Skeletons were prepared for gross morphology analysis using the McLeod method, as previously described[66]. Briefly, mice were sacrificed, skinned, eviscerated, and fixed in 95% ethanol for one day. Then, skeletons were stained by alizarin red s and alcian blue solutions (Sigma, A3157) and sequentially cleared in 1% potassium hydroxide. All images presented are representative of the respective genotypes ($n = 3$–5).

## Histology, immunofluorescence, and immunohistochemistry

Histological analysis was performed on liver and hindlimbs, and skeletons and skeletal muscles were dissected from mice fixed in 4% PFA for one day, and decalcified by daily changes of 15% tetrasodium EDTA for one to two weeks. Tissues were dehydrated by passage through an ethanol series, cleared twice in xylene, embedded in paraffin, and sectioned at 5 μm thickness along the coronal plate from anterior to posterior. Decalcified femoral sections were stained with hematoxylin

and eosin (H&E) or safranin O. Alternatively, intact hindlimbs containing bones and skeletal muscles were immediately fixed in ice-cold 4% paraformaldehyde solution for one day for histological analysis of skeletal muscles. Semi-decalcification was carried out for five days in 0.5 M EDTA pH 7.4 at 4 °C with constant shaking (age ≥1 week), and infiltration was followed with a mixture of 20% sucrose phosphate buffer for one day. All samples were embedded in a 50/50 mixture of 25% sucrose solution and OCT compound (Sakura) and cut into 10 μm thick cross sections using a cryostat (Leica Microsystems, USA).

Immunofluorescence staining and analysis were performed, as described previously[8]. Briefly, after treatment with 0.2% Triton X-100 for 10 min, cryosections were blocked with 5% donkey serum at room temperature for 30 min and incubated overnight at 4 °C with primary antibody and then visualized with fluorescence-conjugated secondary antibody (1:400, Molecular Probes). Nuclei were counterstained with 4–6,diamidino-2-phenylindole (DAPI). An Olympus IX81 confocal microscope or Leica TCS SP5 II Zeiss LSM-880 confocal microscope or EVOS epifluorescence microscope was used to image samples. Osteocalcin (OCN, sc-365797, Santa Cruz Biotechnology, 1:100), ECSIT (HPA042979, Sigma,1:100), ECSIT (gifted from Dr. Sankar Gosh, Columbia University), NDUFS3 (43–9200, Invitrogen,1:100), CD31/PECAM-1 (AF3628, R&D systems, 1:50), Endomucin (sc-65495, Santa Cruz Biotechnology, 1:100), COL1A1 (A1352, ABclonal, 1:50), MF20 (AB_2147781, DSHB, 1:100) and anti-FLAG-tag (MA1-142-A488, Sigma-Aldrich, 1:100) were used for primary antibodies.

For immunohistochemistry, paraffin sections were dewaxed and stained according to the manufacturer's directions, using the Discovery XT automated IHC stainer (Ventana Medical Systems, Inc., Tucson, AZ, USA). CC1 standard (pH 8.4 buffer contained Tris/Borate/EDTA) and inhibitor D (3% $H_2O_2$, Endogenous peroxidase) were used for antigen retrieval and blocking, respectively. Sections were incubated with antibodies specific to ECSIT (A7804, ABclonal,1:200) for 40 min at 37 °C, and a secondary antibody for 20 min at 37 °C. Subsequently, they were incubated in SA-HRP D for 16 min at 37 °C and then DAB + $H_2O_2$ substrate for 8 min, followed by hematoxylin and bluing reagent counterstain at 37 °C. Reaction buffer (pH 7.6 Tris buffer) was used as washing solution. Stained samples were visualized using an Aperio virtual microscope (Leica Microsystems, USA) and images of the sample were analyzed by the Aperio image scope program (ver. 12.3.2.8013, Leica Microsystems, USA).

## Human myoblast and mouse osteoblast and osteoclast differentiation

Human primary myoblasts 17Ubic were kindly gifted from Dr. Charles P Emerson at UMass Chan Medical school. Informed consent was obtained from the patients who donated tissue for the production of cell lines. IRB protocols were approved by UMass Chan Medical School (H00006581-10 and H00006581-11) and by Kennedy Krieger Institute (B0410080117). For human myoblast differentiation, human primary myoblasts 17Ubic[51] were cultured in growth medium (20% FBS,0.5% CEE and 1.2 mM CaCl₂ in Ham's F10 medium) and plated into a collagen-coated 100 mm Petri-dish. To induce myogenic differentiation and fusion, cells were cultured in growth medium until >95% confluency and changed with differentiation medium (OPTI-MEM). Knockdown of ECSIT expression in 17Ubic cells was performed using lentivirus-mediated delivery of human *ECSIT* shRNA hairpins and knockdown efficiency was confirmed by immunoblotting. Alternatively, conditioned medium was harvested from cultured *Ecsit;*$^{+/}$$^{+}$*Prx1-Cre;Rosa*$^{mT/mG}$ or *Ecsit;*$^{fl/fl}$*Prx1-Cre;Rosa*$^{mT/mG}$ skeletal cells and added to 17Ubic cells. 6 days after myogenic culture (30% conditioned medium + 70% myogenic differentiation medium), myoblast differentiation was analyzed by immunofluorescence for MF20 and RT-PCR for myogenic gene expression. The ALK inhibitor SB-431542 (S4317) was purchased from Sigma-Aldrich and added to the culture at a final concentration of 1 μM.

For mouse osteoblast differentiation, GFP-expressing *Prx1*⁺ skeletal cells were FACS-sorted from P1 *Ecsit;*⁺/⁺*Prx1-Cre;Rosa*^mT/mG^ and *Ecsit;*^fl/fl^*Prx1-Cre;Rosa*^mT/mG^ neonates using cell surface markers (CD45⁻Tie2⁻Ter119⁻). 6 days after osteogenic culture, skeletal cells were fixed with10% neutral formalin buffer and stained with the solution containing Fast Blue (Sigma-Aldrich, FBS25) and Naphthol AS-MX (Sigma-Aldrich, 855) to assess alkaline phosphatase (ALP) activity. Subsequently, cells were washed and incubated with a solution containing 6.5 mM Na₂CO₃, 18.5 mM NaHCO₃, 2 mM MgCl₂, and phosphatase substrate (Sigma-Aldrich, S0942), and ALP activity was measured by spectrometer (BioRad). Finally, cell survival was analyzed by staining with Alamar Blue solution (DAL1100, Invitrogen).

For osteoclast differentiation, bone marrow cells were flushed from long bones of two-month-old *Ecsit*^fl/fl^ mice and cultured in the presence of 10 ng/ml M-CSF (R&D systems) to obtain bone marrow-derived macrophages (BMMs). Twenty-four hours later, the non-adherent cells were re-plated, cultured in the medium containing 20 ng/ml of M-CSF for 3 days, and then 10 ng/ml of RANKL were added to induce osteoclast differentiation. Twelve hours later, BMMs were treated with rAAV9 vectors expressing control vector or Cre recombinase, cultured in the presence of 20 ng/ml of M-CSF and 10 ng/ml of Rank ligand for 6 days, and osteoclast differentiation was analyzed by TRAP staining and RT-PCR analysis.

## Quantitative RT-PCR and immunoblotting

Fresh tibialis anterior (TA) muscle and gastrocnemius (GA) muscle tissues were dissected and homogenized mechanically in QIAzol (QIAGEN) to isolate total RNA samples. cDNA was synthesized using the HighCapacity cDNA Reverse Transcription Kit from Applied Biosystems. Quantitative RT-PCR was performed using SYBR® Green PCR Master Mix (Applied Biosystems) with QuantStudio (TM) 6 Flex System (Applied Biosystems). The primers used for PCR are described in Supplemental Table 2.

For immunoblotting analysis, cells were lysed in TNT lysis buffer (50 mM Tris-HCl (pH 7.4), 150 mM NaCl, 1% Triton X-100, 1 mM EDTA, 1 mM EGTA, 50 Mm NaF, 1 mM Na₃VO₄, 1 mM PMSF, and protease inhibitor cocktail (Sigma-Aldrich), and protein amounts from cell lysates were measured using the DC protein assay (BioRad). Equivalent amounts of proteins were subjected to SDS-PAGE, transferred to Immunobilon-P membranes (Millipore), immunoblotted with antibodies specific to ECSIT (HPA042979, Sigma,1:1000), ECSIT (a gift from Dr. Sankar Gosh, Columbia University), NDUFS3 (43−9200, Invitrogen, 1:1000), PINK1 (BC100, Novus Biologicals, 1:1000), LC3B (2775S, Cell Signaling Technology, 1:1000), ND6 (NBP2−94464, Novus Biologicals, 1:1000), NDUFAF1 (PA5−59051, Thermofisher, 1:1000), HSP90 (4877s, Cell Signaling Technology, 1:1000), RUNX2 (sc-390351, Santa Cruz Biotechnology,1:1000) and developed with ECL (Thermo Fisher Scientific). Immunoblotting with anti-GAPDH antibody (CB1001, Emd millipore, 1:1000) was used as a loading control. Anti-OXPHOS Cocktail Kit was purchased from ThermoScientific (458099, 1:250).

## RNA sequencing and whole transcriptome analysis

RNA-seq analysis was performed using OneStopRNAseq[67]. Specifically, FastQC[68] and MultiQC[69] were used for raw read quality control and QoRTs[70] for post-alignment quality control. Paired-end reads were aligned to mouse genome mm10, with star_2.7.5a, annotated with gencode.vM25.primary_assembly annotation[71]. Aligned exon fragments with mapping quality higher than 20 were counted toward gene expression with featureCounts_2.0.0[72] with default settings except for the following parameters '-Q 20−minOverlap 1−fracOverlap 0 -p -B -C'. Differential expression (DE) analysis was performed with DESeq2_1.28.1[73]. Within DE analysis, 'ashr' was used to create log2 Fold Change (LFC) shrinkage for each comparison. Significant DE genes (DEGs) were filtered with the criteria FDR <0.05 and absolute log2 fold change (|LFC|) >0.585. Gene set enrichment analysis was performed with GSEA[74].

## Flow cytometry analysis

Bone marrow cells and splenocytes were isolated from p10 *Ecsit*^Prx1^ and *Ecsit*^fl/fl^ mice. After incubation with red blood cells (RBC) lysis buffer containing NH₄Cl, KHCO₃, and EDTA), cells were passed through a 40 μm cell strainer, washed with FACS buffer (cold PBS (pH 7.2) containing 0.5% BSA (Fraction V) and 1 mM EDTA), and stained with mouse hematopoietic lineage antibody Cocktail (a mixture of antibodies against CD3, CD4, CD8, B220, Gr-1, CD11b and Ter119, 88−7772-72, Invitrogen, 1:100) and antibodies for cKit (12-1171-83, Invitrogen, 1:100) and Sca-1 (17-5981-82, Invitrogen, 1:100) for 30 min at 4 °C, washed with PBS, then 7-amino-actinomycin D (7AAD, 00-6993-50, Invitrogen, 1:1000) was added right before flow cytometry analysis to stain dead cells. Alternatively, cells were stained with Gr-1 (11-5931-85, Invitrogen, 1:100), B220 (48-0452-82, Invitrogen, 1:100), CD3ε (17-0031-82, Invitrogen, 1:100), CD11b (12-0112-82, Invitrogen, 1:100) to analyze immune cell populations. After washing three times, cells were re-suspended in cold PBS (pH 7.2) with 1 mM EDTA and analyzed with a LSRII (BD Biosciences) with the exclusion of 7AAD⁺ cells and doublets.

For flow cytometry analysis of muscle satellite cells, skeletal muscle tissues were dissected from P15 *Ecsit*^Prx1^ and *Ecsit*^fl/fl^ hindlimbs and after 1 h enzymatic digestion at 37 °C, cells were stained with the following antibodies. CD45 (103103, BioLegend, 1:100), CD11b (101203, BioLegend, 1:100), Ter119 (116203, BioLegend, 1:100), Sca1 (108143, BioLegend, 1:100), integrin β-1 (102205, BioLegend, 1:100) and CXCR4 (146507, BioLegend, 1:100) for muscle-resident satellite cell staining while CD31 (102503, BioLegend, 1:100), CD45 (103103, BioLegend, 1:100), integrin-α7 (130-123-833, Miltenyi Biotec, 1:100), Sca1(108143, BioLegend, 1:100) for staining of skeletal muscle resident stem/progenitor cells. Ghost Dye (13−0865, Tonbo Biosciences, 1:100) was used to stain dead cells.

For flow cytometry analysis of skeletal stem cells, P1 *Ecsit*^Prx1^ and *Ecsit*^fl/fl^ limbs were dissociated by mechanical and enzymatic digestion (1mg/ml of Collagenase D (11088866001, Sigma-Aldrich), 2 mg/ml of Dispase II (4942078001, Roche), 1 mg/ml of Hyaluronidase (H3506, Sigma-Aldrich) and 10000 unit/ml of DNase I (4716728001, Roche) for 1 hour at 37 °C under gentle agitation. After blocking with purified rat anti-mouse CD16/CD32 (101302, BioLegend, 1:100) for 1 hour on ice, cells were stained with Biotin-conjugated CD45 (103103, BioLegend, 1:100), Tie2 (124005, BioLegend, 1:100) and Ter119 (116203, BioLegend, 1:100) with BV421-conjuated streptavidin (405226, BioLegend, 1:100), PE-conjugated CD51(104105, BioLegend, 1:100), BV605-conjugated Thy1.1-2 (140317, BioLegend, 1:100), APC-conjugated CD200 (123809, BioLegend, 1:100), FITC-conjugated LY51 (108305, BioLegend, 1:100) and PE/Cy7-conjugated CD105 (120409, BioLegend, 1:100) for 45 min on ice. The grating strategy of flow cytometer to analyze skeletal progenitors is described in Supplemental Fig. 3. Alternatively, cells were isolated from P1 *Ecsit;*⁺/⁺*Prx1-Cre;Rosa*^mT/mG^, and *Ecsit;*^fl/fl^*Prx1-Cre;Rosa*^mT/mG^ limbs and stained with CD45, Tie2, and Ter119. After washing three times, cells were re-suspended in cold PBS (pH 7.2) with 1 mM EDTA and 1 μg/ml DAPI and sorted with a FACSAria II SORP cell sorter (Becton Dickinson) with exclusion of DAPI⁺ cells and doublets.

## Bone fracture

Bone fracture surgery was performed on 10-week-old male *Osx-cre*, *Ecsit*^Osx^, and *Ecsit*^fl/fl^ mice. A longitudinal skin incision was made along the lateral aspect of the thigh from the stifle joint to the hip. The lateral aspect of the femur was exposed by parting the vastus lateralis muscle and the rectus femoris muscle to expose the length of the femur while preserving the femoral nerve. The middle of the femoral shaft was excised with a surgical saw. Intramedullary fixation was performed with a 25G needle penetrating from the patella furrow of the distal femur to the greater trochanter tip of the femur. Both ends of the needle were bent and then cut with a wire cutter, leaving 1 mm. Fascia was sutured using a 4/0 Vicryl suture, and then skin was closed using a

4/0 Nylon suture. Radiographs of the injured legs were performed to monitor fracture healing 2 weeks post-surgery. Ten weeks later, microCT and histology were performed for skeletal analyses. Alternatively, 6 week-old *Prx1;Rosa26^mTmG* mice, while still under surgical plane of anesthesia, were placed into an Einhorn Device to create a closed femoral fracture[75], which was then confirmed by radiography. Nonunion fracture was defined as failure of bridging callus on anteroposterior and lateral fractured cortical bones by coronal and sagittal reconstruction view of microCT and radiography. Fracture unionization was defined as osseous consolidation evident on reconstruction view of microCT and radiography.

### AAV-mediated expression of ECSIT proteins in mice
A single dose of rAAV9 vectors carrying control vector or FLAG-ECSIT constructs ($2 \times 10^{11}$ GC, 50 µl) was randomly injected into P1 *Ecsit^fl/fl* and *Ecsit^Prx1* neonates via the facial vein and 21 days later, skeletal and muscular phenotypes were assessed using X-ray, microCT, and histology. Alternatively, a single dose of rAAV9 vector carrying mCherry ($2 \times 10^{11}$ GC, 50 µl) was injected into P1 *Prx1;Rosa26^mTmG* neonates via facial vein. Twenty-one days later, mCherry expression in individual tissues and the cryosectioned bone and its adjacent muscle were monitored by the IVIS-100 optical imaging and fluorescence microscopy, respectively.

### Mitochondrial fractionation
Human bone marrow-derived stromal cells (BMSCs) were purchased from ScienCell Inc. (Cat #: 7500). Cells were resuspended in mitochondria buffer (70 mM sucrose, 1 mM EGTA, 210 mM sorbitol, 10 mM MOPS [pH 7.4])[41] and then, the supernatant was collected after centrifugation for 10min (450g, 4 °C). For total fraction, 54 ul of the supernatant was collected after centrifugation for 10 min (550 g, 4 °C). The rest of supernatant was further centrifuged at 9500g for 10min at 4 °C and then, the supernatant was collected for the cytosol fraction. For mitochondria fraction, the pellet was resuspended in mitochondrial buffer, incubated on ice for 15 min, and centrifuged at 500 g for 10 min at 4 °C. The supernatant was further centrifuged at 9500g for 10min at 4 °C. Finally, the pellet was collected and resuspended in 54 ul mitochondria buffer as mitochondria fraction.

### Statistical analysis
All data were presented as the mean ± SD. Sample sizes were calculated on the assumption that a 30% difference in the parameters measured would be considered biologically significant with an estimate of sigma of 10–20% of the expected mean. Alpha and Beta were set to the standard values of .05 and 0.8, respectively. No animals or samples were excluded from analysis, and animals were randomized to treatment versus control groups, where applicable. For relevant data analysis, where relevant, we first performed the Shapiro-Wilk normality test for checking normal distributions of the groups. If normality tests passed, two-tailed, unpaired Student's *t*-test were used, but if normality tests failed, Mann-Whitney tests were used for the comparisons between two groups. For the comparisons of three or four groups, we used one-way ANOVA if normality tests passed, followed by Tukey`s multiple comparison test for all pairs of groups. If normality tests failed, Kruskal–Wallis test was performed and was followed by Dunn`s multiple comparison test. For the comparison of two proportions, we performed Fisher's exact test. The GraphPad PRISM software (v.9.4.1, La Jolla, CA) and Microsoft Office Excel 2016 were used for statistical analysis. ns, not significant; $P < 0.05$ was considered statistically significant. *$P < 0.05$; **$P < 0.01$; ***$P < 0.001$; and ****$P < 0.0001$.

### Reporting summary
Further information on research design is available in the Nature Portfolio Reporting Summary linked to this article.

### Data availability
Data supporting the findings of this manuscript are available from the corresponding author upon request. The raw data are protected and are not available due to data privacy laws. However, the processed data generated in this study are provided in the Supplementary Information and Source Data file. In addition, the RNA-seq data generated in this study has been deposited in the NCBI database under accession code GSE214454. Source data are provided with this paper.

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

## Acknowledgements

We would like to thank many individuals who provided reagents and the Flow Core for FACS sorting (S10 grant 1S10OD028576). This project was supported by a NIH NIAMS grant: R21AR077557 and AAVAA Therapeutics. M.B.G. holds a Career Award for Medical Scientists from the Burroughs Wellcome Fund, NIH grant: R01AR075585, and a Pershing Square Sohn Cancer Research Alliance award.

## Author contributions

C.L. designed, executed, and interpreted the experiments. J.X. and G.G. generated the AAV vectors. Q.Y. and D.G. performed mitochondrial and skeletal muscle experiments, respectively. Y.Y., Z.C., and A.A.J. performed histology and microCT. S.C. N.D. and Q.Q. performed flow cytometer analysis of bone marrow. S.C. performed osteoclast experiments. W.T.O. performed bone fracture experiments. R.L. and L.J.Z. performed whole transcriptome analysis. S.G. generated Ecsit-floxed mice. M.G., S.L., C.H., and C.P.E. interpreted the experiments and helped draft the manuscript. J.H.S. supervised the research and prepared the manuscript.

## Competing interests

G.G. and J.H.S. are co-founders of AAVAA Therapeutics and hold equity of this company. G.G. is a scientific co-founder of Voyager Therapeutics and Aspa Therapeutics Inc., and an inventor on patents with potential royalties licensed to other biopharmaceutical companies, in which they hold equity. Other authors declare no competing interests.
