## [Peer Review File · Nature Communications]

Impaired mitochondrial oxidative metabolism in skeletal progenitor cells leads to musculoskeletal disintegrationREVIEWER COMMENTS

Reviewer #1 (Remarks to the Author):

This study provides in vivo evidence of importance of mitochondrial oxidative metabolism in determining developmental bone formation. The proposed mechanism is that mitochondrial oxidative metabolism is critical for the osteogenic commitment and differentiation of skeletal progenitors.

Authors have deleted Ecsit in skeletal progenitors using Prx1Cre (EcsitPrx1) resulting in almost no bone formation and regeneration capability, skeletal deformity, defects in the bone marrow niche, and spontaneous fractures followed nonunion. In contrast to this phenotype practically no bone phenotype was observed when using targeted deletion to osteoprogenitors using SP7-Cre and to more mature osteoblasts using DMP1Cre. Also, no phenotype when deletion is induced in osteoclasts using CatKCre.

In addition, deletion using Prx-1Cre results in muscle phenotype as a result of invasion of skeletal progenitors and postulated to be from increase in TGF β -1 production by SSCs (progenitors in bone).

This paper is interesting but has some major shortcomings critical to support the hypothesis that this mechanism is via effects on skeletal progenitor cells SSPs rather than on the osteoblasts that are major energy consumer in the organism.

- To prove that PRXCre targeting of skeletal SSPs is critical for the effects, and not targeting osteoprogenitors/osteoblasts/osteocytes in which PRX also, very efficiently deletes ECSIT additional experiments should be done. This includes showing IHC for bone samples as shown in Fig 1a, as it appears that antibody works very well to show presence of ECSIT. To make this comment straightforward similar sections from PrxCre;Ecsitfl/fl, and control versus straining of SP7;Ecsitfl/fl, DMP1Cre;Ecsitfl/fl to show what is the efficiency of deletion using these three promoters using IHC 9 (Fig 1a). Efficiency of recombination (RNA level) should be presented side by side. If recombination is much more efficient using PRXCre that the option of effects via targeting osteoblasts with Prx promoter can't be excluded as a cause of phenotype.
- To show the effects of deletion during fracture repair, an inducible system should be utilized. Doing fractures in bones that are severely affected during development does not show the clear effects on progenitors during fracture healing as pre-existing phenotype and developmentally changed progenitor population can result in healing changes.
- AAV delivery rescue can result in effects on osteoprogenitor or mature osteoblasts, cells that are targeted by Prx-1 as well, and not just from progenitor cells. AAV rescue does not exclude that effects are exclusive to SSPs and not from osteoblasts.
- Interesting muscle phenotype and implication of TGF β -1 suggested that SSPs produced TGF β 1 has a role. How to exclude that effects are not from satellite cells derived TGF β 1. Can a Pax7CreER be used to target deletion of TGF β 1.
- CatKCre effects are not present and they do not contribute at all to the premise, and to this manuscript and should be taken out.

Reviewer #2 (Remarks to the Author):

The role of mitochondrial oxidative metabolism in the remodelling of bone and skeletal muscle integrity is largely unexplored. In this study, Lin et al have generated several cell-type specific knockout mouse models of the complex I assembly factor ECSIT to interrogate the role of oxidative metabolism on bone remodelling, healing and skeletal muscle homeostasis. The major findings presented are that the loss of Ecsit in skeletal progenitor cells leads to spontaneous fractures and affects skeletal development and regeneration. The authors have undertaken a very large study here and much of the data is conclusive (and perhaps not unexpected for a protein that regulates complex I assembly). They also nicely established that the findings are due to

Ecsit's location in mitochondria, consistent with other reports that Ecsit is critical for complex I assembly as part of the MCIA complex. Surprisingly however, the authors also demonstrated that loss of Ecsit in committed skeletal cells did not lead to defects in bone morphology, fracture healing and osteoclast differentiation and complex I (using a subunit as surrogate) is expressed. Finally, the authors show that mice lacking Ecsit in skeletal progenitor cells develop myopathy and muscle atrophy, and that either expression of Ecsit using rAAV or inhibition of the myogenic suppressor TGF- β 1 was able to rescue many of the defects observed.

While this study provides new insights into the interplay of mitochondrial oxidative phosphorylation (through complex I) and the function of cells associated with bone remodelling, I have a number of concerns that need to be addressed. The main concern is the conclusion that Ecsit is redundant for complex I assembly in some cell types. In my opinion, this conclusion requires further validation given that loss of Ecsit leads to loss of two other assembly factors of the MCIA (NDUFAF1 and TMEM126B), which are also required complex I assembly.

Major comments:

1. The data presented in Figure 4 suggests that Ecsit is not required for complex I assembly in Osteoprogenitor, Osteoclasts or Osteoblasts (Fig. 4G). Given that the Ecsit mRNA levels are only reduced by ~50% (Fig. 4B, C), the authors should perform Western blot analysis similar to Fig. 4G to establish that Ecsit is indeed missing in osteoprogenitor, osteoblast and osteoclast cells. This is critical since many groups have shown that small amounts of assembly factor are sufficient to rescue complex I assembly in cells. Furthermore, the authors use NDUFS3 as a surrogate for complex I (which is not a strong signal in most cases). Additional complex I subunits should be analysed.
2. Similarly, it has been shown extensively that Ecsit is part of the MCIA complex (Heide et al, 2012; Guerrero-Castillo et al, 2017; Formosa et al, 2020) and knockdown/knockout of Ecsit leads to reduced levels of other components of this complex (Vogel et al, 2007; Nouws et al, 2010; Formosa et al, 2020). The authors should check the stability of Ndufaf1 and/or Tmem126b in these cell types to see if they are destabilized in the absence of Ecsit. This should also be performed in shECSIT experiments in 17Ubic cells (As shown in Fig 5J, K).
3. The specificity of the Ecsit antibody is not clear. The authors attempt to show colocalization between Ecsit and NDUFS3 in Fig 1, yet this is not at all clear – e.g. in panel 1C Ecsit shows a unequal distribution and not a clear mitochondrial pattern. The authors show that Ecsit is mainly in mitochondria based on western blot analysis which is consistent with other studies. They should show a whole range of the blot to ensure that there are not non-specific proteins that obscure the results.
4. The authors show in Figure 3D that the expression of Ecsit- Δ MLS survival relative to the VEC control. How? Furthermore, Ecsit- Δ MLS appears to reduce TGF- β 1 transcripts by ~50% relative to the cKO treated with VEC (Fig. 6I). The authors should establish that Ecsit- Δ MLS is not found in mitochondria.
5. In Fig. 2C, what antibody was used as a surrogate for CI given that NDUFS3 is also used? Also, given that these are looking at subunits on SDS-PAGE rather than assembled OXPHOS complexes, this should be clarified for all.

Minor comments:

6. The schematic in Fig 3A should have 'AAV-ECSIT- Δ MLS' not 'AAV-ECSIT-MSL'. Similarly for 3F, 'MSL' should be 'MLS'
7. In the schematic in the extended data Fig 9, the electron flow is not accurate (IMS transfer by cyt c should be shown) and the oxphos complexes should transit the membrane. Also ecsit faces the matrix not the IMS.

Reviewer #3 (Remarks to the Author):

In this manuscript, Lin and colleagues examine the function of the Ecsit protein in skeletal development. Ecsit contributes to the stabilization of mitochondrial complex I and is therefore critical to normal cellular metabolism. The authors demonstrate that Ecsit ablation in Prx1+ cells leads to severe skeletal malformations. The mutant mice develop spontaneous fractures, impaired ossification center formation, and impaired fracture healing. While the phenotypes of the mouse models are quite astounding, the manuscript lacks mechanistic depth. More substantial and in-depth analyses of the Prx1-Cre mediated knockout are necessary.

- 1) The primary defect in the Ecsit deficient Prx1+ is not clear to me. The authors suggest that mitochondrial numbers are reduced but expression of CII, CIII, CIV, and CV proteins are normal. These data would appear to be incompatible. Is mitochondrial biogenesis effected or mitophagy induced.
- 2) The authors show data on oxygen consumption rate, but what about ECAR or another measure of glycolysis. It might be expected that the function of this metabolic pathway is heightened, and if so how does it influence ATP levels. Does energetic stress impinge on the function of RUNX2 and thereby prevents normal osteoblast development.
- 3) It is difficult to remedy the notion that Ecsit has an essential function in Prx1+ cells but is dispensable in Osx+ cells and more mature osteoblasts given the increase in mitochondrial number and activity during differentiation.
- 4) Additional quantitative analysis of the fracture healing studies is necessary.

Reviewer #1 (Remarks to the Author):

This study provides in vivo evidence of importance of mitochondrial oxidative metabolism in determining developmental bone formation. The proposed mechanism is that mitochondrial oxidative metabolism is critical for the osteogenic commitment and differentiation of skeletal progenitors. Authors have deleted Ecsit in skeletal progenitors using Prx1Cre (EcsitPrx1) resulting in almost no bone formation and regeneration capability, skeletal deformity, defects in the bone marrow niche, and spontaneous fractures followed nonunion. In contrast to this phenotype practically no bone phenotype was observed when using targeted deletion to osteoprogenitors using SP7-Cre and to more mature osteoblasts using DMP1Cre. Also, no phenotype when deletion is induced in osteoclasts using CatKCre. In addition, deletion using Prx-1Cre results in muscle phenotype as a result of invasion of skeletal progenitors and postulated to be from increase in TGF β -1 production by SSCs (progenitors in bone). This paper is interesting but has some major shortcomings critical to support the hypothesis that this mechanism is via effects on skeletal progenitor cells (SSPs) rather than on the osteoblasts that are major energy consumer in the organism.

- We thank the reviewer for summarizing and highlighting the significance of our manuscript. We appreciate the reviewer's constructive suggestions and believe that addressing these points has strengthened the manuscript.

- To prove that PRXCre targeting of skeletal SSPs is critical for the effects, and not targeting osteoprogenitors/osteoblasts/osteocytes in which PRX also, very efficiently deletes ECSIT additional experiments should be done. This includes showing IHC for bone samples as shown in Fig 1a, as it appears that antibody works very well to show presence of ECSIT. To make this comment straightforward, similar sections from PrxCre;Ecsitfl/fl, and control versus straining of SP7;Ecsitfl/fl, DMP1Cre;Ecsitfl/fl to show what is the efficiency of deletion using these three promoters using IHC (Fig 1a). Efficiency of recombination (RNA level) should be presented side by side. If recombination is much more efficient using PRXCre that the option of effects via targeting osteoblasts with Prx promoter can't be excluded as a cause of phenotype.*

- As suggested by the reviewer, protein levels of Ecsit in osteoprogenitors, osteoblasts, and/or osteocytes were assessed using immunohistochemistry (IHC) for ECSIT on longitudinal sections of *Osx-cre*, *Ecsit^{Osx}*, *Ecsit^{fl/fl}*, and *Ecsit^{Dmp1}* femurs, respectively (**Fig. 4b, d, right panels**). Additionally, mRNA levels of *Ecsit* in the tibial bones isolated from same mice were examined by RT-PCR (**Fig. 4b, d, left panels**). Finally, protein levels of Ecsit in calvarial osteoblasts (COBs) isolated from *Osx-cre* and *Ecsit^{Osx}* neonates or in bone marrow-derived stromal cells (BMSCs) isolated from the long bones of *Ecsit^{fl/fl}* and *Ecsit^{Dmp1}* mice were assessed by immunoblot analysis using an anti-Ecsit antibody (**Fig. 4g, middle panels**). These results demonstrated that the deletion efficiency of *Ecsit* using PRX1-Cre, SP7-Cre, or DMP1-Cre deleter strain is comparable, suggesting that deletion of *Ecsit* in cells in the osteoblast differentiation sequence prior to the expression of SP7 or DMP1 is responsible for the skeletal phenotypes observed.

- *To show the effects of deletion during fracture repair, an inducible system should be utilized. Doing fractures in bones that are severely affected during development does not show the clear effects on progenitors during fracture healing as pre-existing phenotype and developmentally changed progenitor population can result in healing changes.*

- We thank the reviewer for raising this interesting point. *Ecsit^{fl/fl}* mice were crossed with *PRX1-CRE^{ERT2}-EGFP* mice (*Ecsit^{Prx1-ERT/GFP}*, Jackson laboratory) to delete *Ecsit* expression in skeletal progenitors by tamoxifen treatment. GFP expression was used to monitor tamoxifen-induced expression of CRE recombinase in *Prx1⁺* osteoblast-lineage cells. Fracture surgery was performed on the right femurs of 6-week-old *Ecsit^{Prx1-ERT/GFP}* and *Prx1-ERT/GFP* (control) mice three days after five consecutive intraperitoneal injection with tamoxifen (**Extended Data Fig. 4a**). Left femurs were used as a non-fracture control to examine basal bone mass in these mice. 32 days after the surgery, tamoxifen-induced expression of PRX1-CRE recombinase and deletion efficiency of *Ecsit* in the tibia were examined using EGFP expression (fluorescence microscopy) and *Ecsit* mRNA expression (RT-PCR), respectively (**Extended Data Fig. 4b, c**). MicroCT analysis revealed that basal bone mass in control and *Ecsit^{Prx1-ERT/GFP}* femurs were comparable, as shown by equivalent trabecular bone

mass and cortical bone thickness of non-fractured femurs (**Extended Data Fig. 4d, e**). Unlike *Ecsit*^{Prx1} mice, tamoxifen-treated *Ecsit*^{Prx1-ERT/GFP} mice displayed normal periosteal and callus formation and fracture unionization in the fractured sites (**Extended Data Fig. 4b, f, g**). These results demonstrate that inducible deletion of *Ecsit* in *Prx1*⁺ skeletal progenitors at adult stage does not affect bone fracture healing and homeostasis. Thus, impaired fracture healing seen in *Ecsit*^{Prx1} mice may result from pre-existing skeletal phenotypes and/or alteration of skeletal progenitor population during early skeletal development. However, it cannot be fully excluded that inadequately complete deletion of *Ecsit* in this system contribute to the lack of a phenotype observed.

- *AAV delivery rescue can result in effects on osteoprogenitor or mature osteoblasts, cells that are targeted by Prx-1 as well, and not just from progenitor cells. AAV rescue does not exclude that effects are exclusive to SSPs and not from osteoblasts.*

- We thank the reviewer for correcting this. We have revised that systemically delivered AAVs can transduce all *Prx1*⁺ osteoblast-lineage cells, including skeletal progenitors, osteoprogenitors, and mature osteoblasts, and that AAV-mediated expression of Flag-tagged ECSIT proteins in these cells can rescue *Ecsit*^{Prx1} skeletal phenotypes.

- *Interesting muscle phenotype and implication of TGFβ-1 suggested that SSPs produced TGFβ1 has a role. How to exclude that effects are not from satellite cells derived TGFβ1. Can a Pax7CreER be used to target deletion of TGFβ1.*

- We thank the reviewer for raising this interesting point. Since *Pax7-CRE^{ERT}* mice are not available from any commercial sectors (Jackson laboratory, Taconic, Charles River) and our collaborators, as an alternative approach, satellite cells were sorted from the skeletal muscles of *Ecsit*^{fl/fl} and *Ecsit*^{Prx1} mice using cell surface markers (β -integrin⁺CXCR4⁺Sca1⁻CD45⁻CD11b⁻TER119⁻) and TGFβ1 expression in these cells was examined by RT-PCR. This demonstrated that mRNA levels of TGFβ1 were comparable between *Ecsit*^{fl/fl} and *Ecsit*^{Prx1} satellite cells (**Fig. 6f**). Similarly, *Ecsit*-deficient myoblast

precursors (17Ubc cells) showed normal TGF β 1 expression (**Fig. 6g**). Thus, *Prx1*+ skeletal cells, not satellite cells, are likely to contribute to the effects of altered TGF β 1 expression in the skeletal muscle of *Ecsit*^{*Prx1*} mice. Additionally, *Ecsit* deletion in myoblast-lineage cells does not affect TGF β 1 expression in this lineage.

- *CatKCre* effects are not present and they do not contribute at all to the premise, and to this manuscript and should be taken out.

- As suggested by the reviewer, the results of *Ecsit*^{*Ctsk*} mice and *Ecsit*-deficient osteoclasts were removed from the revised manuscript.

Reviewer #2 (Remarks to the Author):

The role of mitochondrial oxidative metabolism in the remodeling of bone and skeletal muscle integrity is largely unexplored. In this study, Lin et al have generated several cell-type specific knockout mouse models of the complex I assembly factor ECSIT to interrogate the role of oxidative metabolism on bone remodeling, healing and skeletal muscle homeostasis. The major findings presented are that the loss of Ecsit in skeletal progenitor cells leads to spontaneous fractures and affects skeletal development and regeneration. The authors have undertaken a very large study here and much of the data is conclusive (and perhaps not unexpected for a protein that regulates complex I assembly). They also nicely established that the findings are due to Ecsit's location in mitochondria, consistent with other reports that Ecsit is critical for complex I assembly as part of the MCIA complex. Surprisingly however, the authors also demonstrated that loss of Ecsit in committed skeletal cells did not lead to defects in bone morphology, fracture healing and osteoclast differentiation and complex I (using a subunit as surrogate) is expressed. Finally, the authors show that mice lacking Ecsit in skeletal progenitor cells develop myopathy and muscle atrophy, and that either expression of Ecsit using rAAV or inhibition of the myogenic suppressor TGF- β 1 was able to rescue many of the defects observed. While this study provides new insights into the interplay of mitochondrial oxidative phosphorylation (through complex I) and the function of cells associated with bone remodeling, I have a number of concerns that need to be addressed. The main concern is the conclusion that Ecsit is redundant for complex I assembly in some cell types. In my opinion, this conclusion requires further validation given that loss of Ecsit leads to loss of two other assembly factors of the MCIA (NDUFAF1 and TMEM126B), which are also required complex I assembly.

- We thank the reviewer for summarizing and highlighting the significance of our manuscript. We agree that these findings will new insights into the interplay of mitochondrial oxidative metabolism and musculoskeletal integrity. We believe that addressing these comments has significantly improved the revised manuscript.

Major comments:

1. The data presented in Figure 4 suggests that *Ecsit* is not required for complex I assembly in Osteoprogenitor, Osteoclasts or Osteoblasts (Fig. 4G). Given that the *Ecsit* mRNA levels are only reduced by ~50% (Fig. 4B, C), the authors should perform Western blot analysis similar to Fig. 4G to establish that *Ecsit* is indeed missing in osteoprogenitor, osteoblast and osteoclast cells. This is critical since many groups have shown that small amounts of assembly factor are sufficient to rescue complex I assembly in cells. Furthermore, the authors use NDUFS3 as a surrogate for complex I (which is not a strong signal in most cases). Additional complex I subunits should be analyzed.

- As suggested by the reviewer, western blot analyses were performed to examine the deletion efficiency of *Ecsit* in osteoprogenitors (*Ecsit^{Osx}*) or osteoblasts (*Ecsit^{Dmp1}*), demonstrating a significant decrease in *Ecsit* expression (Fig. 4g). These results were also validated using immunohistochemistry (IHC) for ECSIT on longitudinal sections of *Osx-cre*, *Ecsit^{Osx}*, *Ecsit^{fl/fl}*, and *Ecsit^{Dmp1}* femurs (Fig. 4b, d, right panels). These results support that these cre lines all mediate robust deletion of ECSIT osteoblast-lineage cells. As suggested by the reviewer, expression of additional complex I subunits, ND6 and NDUFB8, in *Ecsit^{Prx1}* skeletal cells was examined by western blot analysis, demonstrating that similar to NDUFS3, protein levels of ND6 and NDUFB8 were markedly decreased in these cells (Fig. 2c). Of note, following the suggestion of reviewer 1, studies of *Ecsit*-deficient osteoclasts were removed from the revised manuscript.

2. Similarly, it has been shown extensively that *Ecsit* is part of the MCIA complex (Heide et al, 2012; Guerrero-Castillo et al, 2017; Formosa et al, 2020) and knockdown/knockout of *Ecsit* leads to reduced levels of other components of this complex (Vogel et al, 2007; Nouws et al, 2010; Formosa et al, 2020). The authors should check the stability of *Ndufaf1* and/or *Tmem126b* in these cell types to see if they are destabilized in the absence of *Ecsit*. This should also be performed in *shECSIT* experiments in 17Ubic cells (As shown in Fig 5J, K).

- As suggested by the reviewer, expression of NDUFAF1 in *Ecsit^{Prx1}* skeletal cells (Fig. 2c) or *Ecsit*-deficient 17Ubic cells (Fig. 5k) was examined by western blot analysis, demonstrating that protein levels of NDUFAF1 were markedly reduced in *Ecsit^{Prx1}* skeletal cells while *Ecsit*-deficient 17Ubic cells

showed a mild reduction. These results suggest that deletion of *Ecsit* in skeletal progenitors, not myoblast precursors, leads to destabilization of other components of the MCIA complex. Since an antibody targeting mouse TMEM126B is not commercially available, its expression in *Ecsit*^{Prx1} skeletal cells was unable to be examined by western blot analysis. The references provided by the reviewer were added to the revised manuscript.

3. *The specificity of the Ecsit antibody is not clear. The authors attempt to show colocalization between Ecsit and NDUFS3 in Fig 1, yet this is not at all clear – e.g. in panel 1C Ecsit shows a unequal distribution and not a clear mitochondrial pattern. The authors show that Ecsit is mainly in mitochondria based on western blot analysis which is consistent with other studies. They should show a whole range of the blot to ensure that there are not non-specific proteins that obscure the results.*

- We agree with the reviewer's concern about the immunofluorescence data (**Fig. 1C**) showing unclear expression patterns of Ecsit in human BMSCs. Since our anti-Ecsit antibody shows higher specificity to mouse Ecsit than human ECSIT (**data not shown**), we performed western blot analyses with other anti-Ecsit antibodies in human BMSCs, demonstrating that anti-Ecsit antibody obtained from Dr. Sankar Gosh, Columbia University, is the most specific for human ECSIT among the available reagents. Thus, immunofluorescence (**Fig. 1c, left**) and western blot (**Fig. 1c, right**) analyses were re-performed using this antibody to confirm the mitochondrial localization of ECSIT in human BMSCs. As suggested by the reviewer, a series of revised western blot images were added to the revised manuscript (**Fig. 1c, right**).

4. *The authors show in Figure 3D that the expression of Ecsit-ΔMLS survival relative to the VEC control. How? Furthermore, Ecsit-ΔMLS appears to reduce TGF-β1 transcripts by ~50% relative to the cKO treated with VEC (Fig. 6I). The authors should establish that Ecsit-ΔMLS is not found in mitochondria.*

- As suggested by the reviewer, Flag-*Ecsit*-expressing *Prx1*⁺ skeletal cells were fractionated into cytosolic supernatants and mitochondrial pellets, and immunoblotted with anti-Flag antibody. These results demonstrated that while Flag-*Ecsit*-FL proteins were mainly located in the mitochondria, the

majority of Flag-Ecsit- Δ MLS mutants were detected in the cytosolic fraction (**Fig. 2j**), indicating the expected requirement of the MLS motif for the mitochondrial localization of ECSIT. Despite its requirement for regulation of mitochondrial OXPHOS in *Prx1*⁺ skeletal progenitors, the Ecsit- Δ MLS mutant can partially rescue survival of *Ecsit*^{*Prx1*} mice and TGF- β 1 expression in *Ecsit*^{*Prx1*} skeletal progenitors, suggesting that ECSIT may have additional functional domains important for survival and TGF- β 1 production. Further studies will be necessary to identify these functional domains in *Prx1*⁺ skeletal progenitors. A brief discussion of this point was added to the discussion section of the revised manuscript.

5. In Fig. 2C, what antibody was used as a surrogate for CI given that NDUFS3 is also used? Also, given that these are looking at subunits on SDS-PAGE rather than assembled OXPHOS complexes, this should be clarified for all.

- We apologize for the lack of detail. In **Fig. 2C**, whole cell lysates of *Ecsit*^{*fl/fl*} and *Ecsit*^{*Prx1*} skeletal cells were subjected to SDS-PAGE gel and immunoblotted with cocktail antibodies that detect subunits of the OXPHOS complexes (CI: NDUFB8, CII: SDHB, CIII: UQCRC2, CIV: Cytochrome C, CV: ATP5A). In addition to the expression of NDUFB8 and NDUFS3 in CI, ND6 expression was also assessed as a CI protein. Representative subunits of each OXPHOS complex I, II, III, IV, and V were clarified in the revised **Fig. 2C**.

Minor comments:

6. The schematic in Fig 3A should have 'AAV-ECSIT- Δ MLS' not 'AAV-ECSIT-MSL'. Similarly for 3F, 'MSL' should be 'MLS'

- They were corrected in the revised manuscript.

7. In the schematic in the extended data Fig 9, the electron flow is not accurate (IMS transfer by cyt c should be shown) and the oxphos complexes should transit the membrane. Also *ecsit* faces the matrix not the IMS.

- As suggested by the reviewer, the schematic diagram was corrected in the revised manuscript
(Extended Data Fig 11).

Reviewer #3 (Remarks to the Author):

In this manuscript, Lin and colleagues examine the function of the Ecsit protein in skeletal development. Ecsit contributes to the stabilization of mitochondrial complex I and is therefore critical to normal cellular metabolism. The authors demonstrate that Ecsit ablation in Prx1+ cells leads to severe skeletal malformations. The mutant mice develop spontaneous fractures, impaired ossification center formation, and impaired fracture healing. While the phenotypes of the mouse models are quite astounding, the manuscript lacks mechanistic depth. More substantial and in-depth analyses of the Prx1-Cre mediated knockout are necessary.

- We thank the reviewer for the overall positive comments and for highlighting the significance of our manuscript. We agree that more mechanistic studies for ECSIT's functions in skeletal progenitors were needed to strengthen the manuscript. New supporting data was added to the revised manuscript and the reviewer's comments were fully addressed.

1) *The primary defect in the Ecsit deficient Prx1+ is not clear to me. The authors suggest that mitochondrial numbers are reduced but expression of CII, CIII, CIV, and CV proteins are normal. These data would appear to be incompatible. Is mitochondrial biogenesis effected or mitophagy induced.*

- We thank the reviewer for pointing this out. Our transcriptome and immunoblotting analyses demonstrated that mRNA and proteins levels of CII, CIII, CIV, and CV complex components were both largely intact in the absence of Ecsit while *Ecsit^{Prx1}* skeletal cells showed a significant reduction in the expression of CI transcripts and proteins (**Fig. 2a-c**). Since ECSIT is a key regulator of the MCIA complex essential for CI assembly (Heide et al, 2012; Guerrero-Castillo et al, 2017; Formosa et al, 2020) and its deletion destabilizes components of the MCIA and CI complexes (Vogel et al, 2007; Nouws et al, 2010; Formosa et al, 2020, **Fig. 2a-c**), impaired CI assembly in *Ecsit^{Prx1}* skeletal cells may result in a significant decrease in mitochondrial numbers and activity while limiting the impact on the stability of CII, CIII, CIV, and CV complexes. Accompanying our transcriptome analysis that show little to no enrichment of genes associated with mitochondrial biogenesis, mRNA and protein levels of Atf5 and Hsp90 were largely intact in the absence of Ecsit (**Fig. 2c, Extended Data Fig. 5a**). Notably,

Ecsit^{Prx1} skeletal cells showed increased expression of Lc3b and Pink1 (**Fig. 2c**), key regulators of mitophagy, suggesting enhanced mitophagy activity. These results suggest that accompanying impaired CI assembly, enhanced mitophagy activity contributes to reduced mitochondria numbers in *Ecsit*^{Prx1} skeletal cells by facilitating the removal of defective mitochondria.

2) *The authors show data on oxygen consumption rate, but what about ECAR or another measure of glycolysis. It might be expected that the function of this metabolic pathway is heightened, and if so how does it influence ATP levels. Does energetic stress impinge on the function of RUNX2 and thereby prevents normal osteoblast development.*

- We thank the reviewer for raising these interesting questions. As suggested, extracellular acidification rate (ECAR) was markedly increased in *Ecsit*^{Prx1} skeletal cells (**Fig 2i**), demonstrating a significant increase in the glycolytic proton efflux rate. Impaired mitochondrial OXPHOS by *Ecsit*-deficiency is likely to enhance the glycolytic rate in *Prx1*⁺ skeletal cells as a compensatory mechanism. However, since mitochondrial OXPHOS is a major source of ATP production, ATP levels in *Ecsit*^{Prx1} skeletal cells were markedly reduced despite the enhanced glycolytic rate (**Fig. 2f**). Of note, expression and transcription activity of Runx2 in *Ecsit*^{Prx1} skeletal cells were largely intact (**Extended Data Fig 5b, c**), suggesting that *Ecsit*-mediated mitochondrial regulation is dispensable for Runx2-mediated osteogenesis.

3) *It is difficult to remedy the notion that Ecsit has an essential function in Prx1+ cells but is dispensable in Osx+ cells and more mature osteoblasts given the increase in mitochondrial number and activity during differentiation.*

- We thank the reviewer for raising this question, as mitochondrial function increases over osteoblast maturation. To exclude the possibility that insufficient deletion of *Ecsit* may mask potential defects in mitochondrial function and osteogenic development of *Ecsit*^{Osx} or *Ecsit*^{Dmp1} osteoblast-lineage cells, mRNA and protein levels of *Ecsit* in these cells were examined by RT-PCR, western blot, and immunohistochemistry analyses (**Fig. 4b, d, g**), demonstrating lack of *Ecsit* expression in

osteoprogenitors or mature osteoblasts. Unlike *Ecsit*^{Prx1} skeletal progenitors, these cells showed normal CI assembly, mitochondrial numbers and activities, and osteogenic differentiation (**Fig. 4g-l, Extended Data Fig. 8a**). Additionally, bone accrual and fracture healing were normal in *Ecsit*^{Osx} or *Ecsit*^{Dmp1} mice (**Fig. 4c, e, f, Extended Data Fig. 7, 8b**). These results suggest that *Ecsit* function is dispensable for mitochondrial OXPHOS and osteogenesis in *Osx*⁺ committed osteoblasts and *Dmp1*⁺ mature osteoblasts and osteocytes. Together with *Ecsit*-deficient myoblast precursors showing normal mitochondrial functions and myogenic differentiation (**Fig. 5h-l**), these results suggest that *Ecsit* function is highly specific to cell types and/or differentiation stages. Further studies will be necessary to determine how *Ecsit* regulates mitochondrial oxidative metabolism and osteogenic development in a context and tissue-dependent manner.

4) Additional quantitative analysis of the fracture healing studies is necessary.

- As suggested by the reviewer, quantitative analyses showing fracture repair of *Ecsit*^{Osx} femurs were added to **Fig. 4f**. Of note, following the suggestion of reviewer 1, studies of *Ecsit*-deficient osteoclasts were removed from the revised manuscript.

REVIEWER COMMENTS

Reviewer #1 (Remarks to the Author):

Dear authors,

My comments have been addressed and additional experiments completed. I have a comment on this statement "Since Pax7-CREERT mice are not available from any commercial sectors (Jackson laboratory, Taconic, Charles River)"

Pax7CreER mice are available from Jax, a catalog number Jax stock 017763

As you made effort with an alternative approach I am fine with your data and summary.

Reviewer #2 (Remarks to the Author):

The authors revised manuscript is much improved from the original submission. However, I still have some concerns:

1. As mentioned previously, even though the level of Ecsit are significantly reduced, only small amount of complex I assembly factors are required to assemble complex I. While the levels of Ecsit in osteoprogenitor cells appears to be absent, there still appears to be Ecsit present in the osteoblasts (Fig. 4g). This could have implications in the interpretation of results due to even low amounts of Ecsit present. The immunoblotting is quite weak for Ecsit as well which could prevent some detection of signal. Also, the SDS-PAGE running profile of Ecsit in osteoblasts looks different to the profile of NDUFS3 and GAPDH. Are all of these samples run on the same gel? While I support the conclusions drawn from the loss of Ecsit in EcsitPrx1 skeletal cells in Figure 2, the immunoblot analysis of ND6 and NDUF8 should be performed on osteoblast and osteoprogenitor cells to substantiate that complex I is not reduced in these other cell types.

2. Again, I agree with the conclusions drawn from Ecsit Prx1 skeletal cells, however, the levels of NDUF1 should also be analysed in osteoblast and osteoprogenitor cells, as it is in these cells that the authors suggest that Ecsit is dispensable for complex I assembly.

3. Unfortunately the immunofluorescence in Fig 1c is too saturated and the degree of colocalization between NDUFS3 and ECSIT is poor. I am not convinced by this staining – colocalization should be 100%.

4. It should be noted that the complex IV subunit in the OXPHOS cocktail is not cytochrome c, but Cytochrome c oxidase subunit 1 (COX1).

Reviewer #4 (Remarks to the Author):

The authors provide mostly satisfactory responses to the comments by Reviewer #3.

Additional comment: Ecsit KO appears to eliminate mitochondria themselves but not just OXPHOS (based on the mitotracker and OCR data). The authors primarily discuss about consequences of energy deficits in Ecsit deletion, but the reduced mitochondrial mass should cause much more

greater cellular abnormalities, because many TCA cycle metabolites, such as acetyl-CoA and aKG, have direct roles in lipid synthesis, protein acetylation, histone demethylation, etc. The authors should acknowledge that the phenotype could be also caused by loss of different functions of mitochondria other than ATP synthesis.

Reviewer #1 (Remarks to the Author):

My comments have been addressed and additional experiments completed. I have a comment on this statement "Since Pax7-CREERT mice are not available from any commercial sectors (Jackson laboratory, Taconic, Charles River)" Pax7CreER mice are available from Jax, a catalog number Jax stock 017763. As you made effort with an alternative approach I am fine with your data and summary.

- We thank the reviewer for correcting this. This mouse model may be useful for our future studies.

Reviewer #2 (Remarks to the Author):

The authors revised manuscript is much improved from the original submission. However, I still have some concerns:

- We agree with the reviewer's concerns, which are fully addressed in the revised manuscript.

1. As mentioned previously, even though the level of Ecsit are significantly reduced, only small amount of complex I assembly factors are required to assemble complex I. While the levels of Ecsit in osteoprogenitor cells appears to be absent, there still appears to be Ecsit present in the osteoblasts (Fig. 4g). This could have implications in the interpretation of results due to even low amounts of Ecsit present. The immunoblotting is quite weak for Ecsit as well which could prevent some detection of signal.

- Thank the reviewer for pointing this out. To clarify the deletion efficiency of Ecsit in osteoprogenitors (Ecsit^{Osx}) and osteoblasts (Ecsit^{Dmp1}), immunoblotting images with more robust signal were added to **revised Fig. 4g**. Additionally, band intensities of ECSIT or NDUFS3 from three individual immunoblots were measured with the Image J software and normalized to the housekeeping protein GAPDH. This demonstrated that while the degree of decrease in ECSIT expression between skeletal progenitors (Ecsit^{Prx1}), osteoprogenitors (Ecsit^{Osx}), and osteoblasts (Ecsit^{Dmp1}) was similar, only Ecsit^{Prx1} cells showed a significant decrease in NDUFS3 expression (**Supplemental Fig. 9**). This, we agree with the reviewer's point that, "However, it cannot be fully excluded that inadequately complete deletion of Ecsit in osteoprogenitors or osteoblasts may contribute to the lack of a phenotype observed." was added to the result section of the revised manuscript.

Also, the SDS-PAGE running profile of Ecsit in osteoblasts looks different to the profile of NDUFS3 and GAPDH. Are all of these samples run on the same gel?

- No, these samples were run on different gels. To clarify this, additional immunoblots for GAPDH on the same gel as ECSIT were added to the **revised Fig. 4g**.

While I support the conclusions drawn from the loss of Ecsit in EcsitPrx1 skeletal cells in Figure 2, the immunoblot analysis of ND6 and NDUFB8 should be performed on osteoblast and osteoprogenitor cells to substantiate that complex I is not reduced in these other cell types.

- As suggested by the reviewer, new immunoblot results showing the expression of ND6 or NDUFB8 in osteoprogenitors (Ecsit^{Osx}) and osteoblasts (Ecsit^{Dmp1}) were added to **revised Fig. 4g**, demonstrating little to no decrease in these cells.

2. Again, I agree with the conclusions drawn from Ecsit Prx1 skeletal cells, however, the levels of NDUFAF1 should also be analysed in osteoblast and osteoprogenitor cells, as it is in these cells that the authors suggest that Ecsit is dispensable for complex I assembly.

- As suggested by the reviewer, new immunoblot result showing the expression of NDUFAF1 in osteoprogenitors (Ecsit^{Osx}) and osteoblasts (Ecsit^{Dmp1}) were added to **revised Fig. 4g**, demonstrating little to no decrease in these cells.

3. Unfortunately the immunofluorescence in Fig 1c is too saturated and the degree of colocalization between NDUFS3 and ECSIT is poor. I am not convinced by this staining – colocalization should be 100%.

- We agree with the reviewer's concerns that the immunofluorescence data displays some differences versus the corresponding immunoblotting data (**Fig. 1c, left**). Despite extensive attempts, we are unable to resolve this issue, likely due to limitations regarding the available anti-ECSIT antibody. Thus, taking the reviewer's comment into account, we have decided to remove this immunofluorescence data

from the revised manuscript.

4. *It should be noted that the complex IV subunit in the OXPHOS cocktail is not cytochrome c, but Cytochrome c oxidase subunit 1 (COX1).*

- We thank the reviewer for correcting this.

Reviewer #4 (Remarks to the Author):

The authors provide mostly satisfactory responses to the comments by Reviewer #3.

Additional comment: Escit KO appears to eliminate mitochondria themselves but not just OXPHOS (based on the mitotracker and OCR data). The authors primarily discuss about consequences of energy deficits in Escit deletion, but the reduced mitochondrial mass should cause much more greater cellular abnormalities, because many TCA cycle metabolites, such as acetyl-CoA and aKG, have direct roles in lipid synthesis, protein acetylation, histone demethylation, etc. The authors should acknowledge that the phenotype could be also caused by loss of different functions of mitochondria other than ATP synthesis.

- We agree with the reviewer's comment and have added the suggested point to the discussion of the revised manuscript.

REVIEWERS' COMMENTS

Reviewer #2 (Remarks to the Author):

Thanks to the authors for making these changes - i think it is now acceptable for publication

REVIEWERS' COMMENTS

Reviewer #2 (Remarks to the Author):

Thanks to the authors for making these changes - i think it is now acceptable for publication :

Completed